# LESS IS MORE: EXTREME GRADIENT BOOST RANK-1 ADAPTION FOR EFFICIENT FINETUNING OF LLMS

## ABSTRACT

Fine-tuning Large Language Models (LLMs) has become a crucial technique for adapting pre-trained models to downstream tasks. However, the enormous size of LLMs poses significant challenges in terms of computational complexity and resource requirements. Low-Rank Adaptation (LoRA) has emerged as a promising solution. However, there exists a gap between the practical performance of low-rank adaptations and its theoretical optimum. In this work, we propose eXtreme Gradient Boosting LoRA (XGBLoRA), a novel framework that bridges this gap by leveraging the power of ensemble learning. Inspired by gradient boosting, XGBLoRA iteratively learns and merges a sequence of LoRA adaptations to refine model predictions. It achieves better performance than the standard LoRA, while enjoying the computational efficiency of rank-1 adaptations. We provide theoretical analysis to show the convergence and optimality of our approach, and conduct extensive experiments on a range of natural language processing tasks. The results demonstrate that XGBLoRA consistently outperforms standard LoRA and achieves performance comparable to full fine-tuning with significantly fewer trainable parameters. This work advances parameter-efficient fine-tuning for LLMs, and offers a promising solution for adapting LLMs to downstream tasks while optimizing performance and efficiency.

## 1 INTRODUCTION

Large language models (LLMs) have achieved remarkable success in various natural language processing tasks, enabling breakthroughs in language understanding, generation, and reasoning (Devlin et al., 2019; Radford et al., 2019; Brown et al., 2020b). These models are typically pre-trained on vast amounts of unlabeled text data, and then fine-tuned on specific downstream tasks to adapt their knowledge to the target domain (Wang et al., 2018; Rajpurkar et al., 2016; Williams et al., 2018). However, the enormous size of LLMs, often reaching billions of parameters, poses significant challenges in terms of computational complexity and resource requirements during fine-tuning (Kaplan et al., 2020; Brown et al., 2020b).

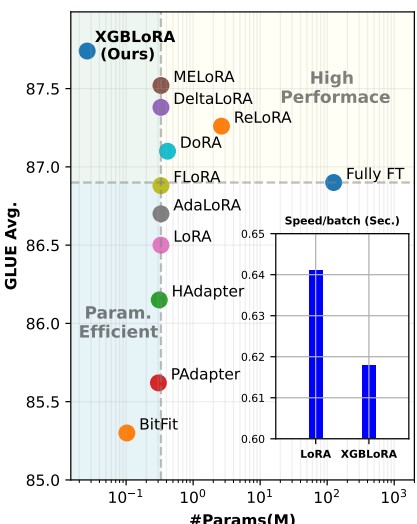

Figure 1: Efficiency *vs.* effectiveness on the GLUE dataset. Our XGBLoRA enjoys high average and uses fewer parameters than competitors. *Mini-figure:* speed in seconds per batch.

To address these challenges, a promising direction called parameter-efficient fine-tuning (PEFT) (Houlsby et al., 2019a; Zaken et al., 2021; Hu et al., 2021) adapts LLMs to downstream tasks while minimizing the number of trainable parameters, thereby reducing the computational and memory overhead. Among these methods, Low-Rank Adaptation (LoRA) (Hu et al., 2021) has gained significant attention due to its effectiveness and simplicity. LoRA freezes the pre-trained model's weights and introduces low-rank matrices to adapt the model to new tasks. By only updating these low-rank matrices during fine-tuning, LoRA significantly reduces the number of trainable parameters compared to full fine-tuning.

Despite its success, LoRA faces a fundamental dilemma between efficiency and effectiveness. (Hu et al., 2021). To guarantee the ability to fit any target matrix, the rank of the adaptation matrices must satisfy the condition of `rank_r` $\geq$ `embedding_size`$/2$. However, in practice, much smaller ranks (*e.g.*, $r \in [8, 16]$) are often used to achieve a good *trade-off* between performance and efficiency. This discrepancy between the theoretical optimum and practical usage leads to a theoretical performance gap. While increasing the rank to match the theoretical requirement helps bridge this gap, it increases the memory usage and computational complexity, diminishing the benefits of using LoRA. This raises an important question:

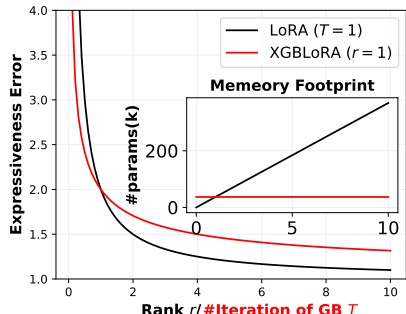

Figure 2: We prove by the error bound in Th. 2 that by compensating for low rank $r = 1$ updates by GB #iterations $T$, XGBLoRA's `err` $\leq C(1 + \frac{1}{\sqrt{T}})$ is close to LoRA's `err` $\leq C(\frac{1}{r} + 1)$. *Mini-figure:* XGBLoRA consumes only $\mathcal{O}(1)$ memory for updates while LoRA consumes $\mathcal{O}(r)$.

*Is there an efficient way to bridge the performance gap while maintaining the extreme low-rank structure and low complexity of LoRA?*

Thus, we propose eXtreme Gradient Boosting Low Rank Adaption (XGBLoRA), a novel framework that resolves the dilemma posed above by leveraging the power of ensemble learning. Inspired by Gradient Boosting (GB) (Friedman, 2001; 2002), XGBLoRA assembles the final model by combining a sequence of boosters (LoRA adaptations), progressively refining the model's predictions. By leveraging the GB principle of *the weak learner* (*i.e.*, strong ensemble model from a set of weak predictors), XGBLoRA lets extreme low-rank adaption overcome the dilemma between efficiency and effectiveness.

Figure 2 illustrates our theoretical analysis: even for rank-1 updates, XGBLoRA can achieve superior performance when combined through gradient boosting, *i.e.*, the expressiveness errors of XGBLoRA and LoRA can remain low while LoRA requires $r\times$ more update memory. Specifically, in Theorems 1 & 2 we establish convergence guarantees and expressiveness bounds that capture the interplay between the number of boosting iterations, the LoRA rank, and the approximation quality. The results reveal that increasing the number of boosting iterations can compensate for lower ranks, letting XGBLoRA bridge the gap between theoretical optimality and practical efficiency.

Through extensive experiments on a range of natural language processing tasks, we demonstrate that XGBLoRA consistently outperforms both standard LoRA and full fine-tuning, while using significantly fewer trainable parameters. For example, in our experiments on LLMs, XGBLoRA runs comfortably on a single NVIDIA RTX 4090 (24GB) for LLaMA3-8B while LoRA requires an A100 (40GB) GPU. Notably, our results show that XGBLoRA with rank-1 updates can match or exceed the performance of higher-rank methods, validating our theoretical insights. Our main contributions are as follows:

i. We introduce XGBLoRA, a novel framework that leverages both ensemble learning and the principle of weak learners for parameter-efficient fine-tuning of large language models. This approach bridges the performance gap between practical usage and theoretical optimum.

ii. We establish convergence guarantees and expressiveness bounds of XGBLoRA in Theorems 1 & 2 that highlight the interplay between the number of GB iterations, the LoRA rank, and the approximation quality. They explain how rank-1 updates achieve superb performance through GB iterations while maintaining low memory update footprint (rank $r$ times lower than LoRA).

iii. Through extensive experiments across a range of NLP tasks, we demonstrate the effectiveness of XGBLoRA which on average achieves better performance than both standard LoRA and full fine-tuning, while using around $10\times$ and $10^4\times$ fewer trainable parameters respectively.

## 2 RELATED WORKS

**Fine-tuning LLMs** has become a prevailing approach for adapting these models to specific downstream tasks (Houlsby et al., 2019b; Diao et al., 2021; Hu et al., 2022; Diao et al., 2023). The process involves training the model on a task-specific dataset, usually with a smaller learning rate compared to pre-training, to adapt its parameters to the target task. Fine-tuning has been successfully applied to a wide range of natural language processing tasks, including text classification, question answering,

and natural language inference (Brown et al., 2020b; Kenton & Toutanova, 2019; Radford et al., 2018; Touvron et al., 2023a). However, fine-tuning LLMs faces several challenges. A major challenge is the computational complexity and memory requirements associated with updating billions of model parameters, which can be prohibitively expensive and time-consuming (Strubell et al., 2019; Brown et al., 2020a; Raffel et al., 2020b; Zhang et al., 2022; Scao et al., 2022; Almazrouei et al., 2023; Touvron et al., 2023a;b; Chiang et al., 2023; Biderman et al., 2023; Jiang et al., 2024). Additionally, the limited availability of labeled data for specific tasks poses challenges in terms of sample efficiency and generalization ability (Zhang et al., 2020). To address these challenges, various approaches have been proposed to improve the efficiency and effectiveness of LLM fine-tuning. One notable technique is LoRA (Hu et al., 2021) which freezes the pre-trained model's weights and introduces a low-rank matrix to adapt the model to new tasks. LoRA reduces the number of trainable parameters and reduces the computational burden of LLM fine-tuning. Recently, (Zeng & Lee, 2024) provided theoretical results that characterize the expressive power of LoRA for Fully Connected Neural Networks (FCNN) and Transformer Networks (TFN), which identify the necessary rank of LoRA for adapting a frozen model to exactly match a target model. For Transformer networks, any model can be adapted to a target model of the same size with LoRA adapters of `rank_r=embedding_size/2`.

**Gradient Boosting** (Friedman, 2001; 2002) is a powerful ensemble learning technique that combines multiple weak learners to create a strong learner. The theoretical foundations of gradient boosting have been extensively studied, providing insights into its convergence properties, generalization ability, and robustness to overfitting. A seminal work on gradient boosting theory by Zhang & Yu (2005) proved that gradient boosting achieves the optimal convergence rate for a broad class of loss functions, highlighting its theoretical optimality. Koltchinskii & Panchenko (2002) further investigated theoretical properties of gradient boosting from the perspective of empirical risk minimization. They derived bounds on the generalization error of gradient boosting and showed that the technique is resilient to overfitting when the base learners are weak and the step size is appropriately chosen. They include insights into its convergence behavior, generalization ability, and robustness, which are relevant to the theoretical analysis of our proposed `XGBLoRA` framework.

## 3 GRADIENT BOOSTING LOW-RANK ADAPTION

### 3.1 PRELIMINARIES

Before delving into the details of `XGBLoRA`, we first introduce key concepts and techniques that form the foundation of our approach. Below, we provide an overview of gradient boosting and LoRA, highlighting their principles, advantages, and relevance to LLM fine-tuning.

**Gradient Boosting (GB)** (Friedman, 2001; 2002) combines multiple weak learners to create a strong learner. The key idea is to iteratively train a sequence of models, each of which corrects mistakes of the preceding model. At each iteration, the model is trained to minimize the residual error between the current predictions and the target outputs. Formally, let $\mathcal{D} = \left\{ (\boldsymbol{x}_i, y_i) \right\}_{i=1}^{N}$ be a dataset of $N$ examples, where $\boldsymbol{x}_i \in \mathbb{R}^d$ is the input feature vector and $y_i \in \mathbb{R}$ is the corresponding target output. The goal of gradient boosting is to learn a function $F(\boldsymbol{x})$ that maps the input features to the target outputs. The function $F(\boldsymbol{x})$ is expressed as a sum of $M$ weak learners $f_m(\boldsymbol{x})$:

$$F(\boldsymbol{x}) = \sum_{m=1}^{M} f_m(\boldsymbol{x}). \tag{1}$$

The weak learners $f_m(\boldsymbol{x})$ are typically simple models, such as decision trees or linear models, that are trained to minimize the residual error. At each iteration $m$, the residual error $r_{im}$ for the $i$-th example is computed as:

$$r_{im} = y_i - F_{m-1}(\boldsymbol{x}_i), \tag{2}$$

where $F_{m-1}(\boldsymbol{x})$ is the cumulative model up to the previous iteration. The weak learner $f_m(\boldsymbol{x})$ is then trained to minimize the loss function $\mathcal{L}_m$ defined over the residual errors:

$$\mathcal{L}_m = \sum_{i=1}^{N} \ell\big(f_m(\boldsymbol{x}_i), r_{im}\big), \tag{3}$$

Figure 3: The pipeline of XGBLoRA: A booster is constructed via randomly choosing $L_s = 2$ adapter layers. Then, it is trained for $\kappa$ steps before merging with the base model. The next booster is then learnt.

where $\ell(\cdot)$ is a differentiable loss function, such as squared error or absolute error. After training the weak learner $f_m(\boldsymbol{x})$, the cumulative model $F_m(\boldsymbol{x})$ is updated as:

$$F_m(\boldsymbol{x}) = F_{m-1}(\boldsymbol{x}) + \alpha_m f_m(\boldsymbol{x}), \tag{4}$$

where $\alpha_m$ is a learning rate that controls the contribution of the weak learner to the final model. The gradient boosting algorithm iteratively repeats this process of computing residual errors, training weak learners, and updating the cumulative model for a fixed number of iterations $M$ or until a convergence criterion is met.

**Low-Rank Adaptation (LoRA)** (Hu et al., 2021) is a PEFT technique designed specifically for adapting large language models to downstream tasks. The key idea behind LoRA is to freeze the pre-trained model's weights and inject trainable low-rank matrices into each layer of the model. By doing so, LoRA significantly reduces the number of trainable parameters while still letting the model adapt to specific tasks. Formally, let $\boldsymbol{W} \in \mathbb{R}^{d \times k}$ be the weight matrix of a fully connected layer in a pre-trained language model, where $d$ is the input dimension and $k$ is the output dimension. LoRA introduces a low-rank decomposition of the weight matrix:

$$\boldsymbol{W} = \boldsymbol{W}_0 + \boldsymbol{AB}, \tag{5}$$

where $\boldsymbol{W}_0$ is the original pre-trained weight matrix, $\boldsymbol{A} \in \mathbb{R}^{d \times r}$ and $\boldsymbol{B} \in \mathbb{R}^{r \times k}$ are trainable matrices, and $r$ is the rank of the adaptation. During fine-tuning, only the matrices $\boldsymbol{A}$ and $\boldsymbol{B}$ are updated, while the original weights $\boldsymbol{W}_0$ remain frozen. This reduces the number of trainable parameters from $dk$ to $(d + k)r$, which is significantly smaller when $r \ll \min(d, k)$. Compared to traditional fine-tuning methods, LoRA offers computational efficiency, memory savings, and efficient model storage, as it eliminates the need to store separate copies of fine-tuned models. Remarkably, despite its parameter efficiency, LoRA has been shown to achieve comparable or even better performance than full fine-tuning on various natural language processing tasks, making it a promising technique for efficient and effective model adaptation. However, LoRA has a theoretical limitation (Zeng & Lee, 2024). To fit any target matrix, the rank of the adaptation must satisfy ($r \geq$ embedding_size$/2$). In practice, much smaller ranks (*e.g.*, $r \in [1, 32]$) are often used to trade-off performance with efficiency. The discrepancy between the theoretical optimum and practical usage leads to a performance gap. Increasing the rank to satisfy the above theoretical requirement increases memory usage and computational complexity, negating the benefits of LoRA by making it as costly as the full fine-tuning strategy.

## 3.2 WHEN GRADIENT BOOSTING MEETS LORA

Below we present the eXtreme Gradient Boosting LoRA method for Transformers, which combines the principles of gradient boosting with the parameter-efficient adaptation technique of LoRA. It iteratively refines a Transformer's predictions by learning a sequence of booster (LoRA adaptations) and merging them into the model's weight matrices. Let $\mathcal{D} = \{(\boldsymbol{x}_i, y_i)\}_{i=1}^{N}$ be a dataset of $N$ examples, where $\boldsymbol{x}_i$ is the input text and $y_i$ is the corresponding target output. Our goal is to fine-tune a pre-trained Transformer $\mathcal{M}_0$ on dataset $\mathcal{D}$ to optimize its performance on the downstream task.

In a Transformer, each layer consists of multiple projection matrices (*e.g.*, query, key, value, output matrices) in the self-attention mechanism, and the weight matrices in the feedforward network. LoRA adapts these matrices by introducing low-rank updates. Specifically, for a weight matrix $\boldsymbol{W} \in \mathbb{R}^{d \times k}$, LoRA performs the following update:

$$\boldsymbol{W} \leftarrow \boldsymbol{W} + \Delta \boldsymbol{W}, \quad \text{where} \quad \Delta \boldsymbol{W} = \boldsymbol{A} \boldsymbol{B}. \tag{6}$$

Here, $\boldsymbol{A} \in \mathbb{R}^{d \times r}$ and $\boldsymbol{B} \in \mathbb{R}^{r \times k}$ are the LoRA matrices, and $r$ is the rank of the adaptation.

The eXtreme Gradient Boosting LoRA for Transformers proceeds in an iterative manner, where at each iteration $t = 1, \ldots, T$, one learns a set of LoRA matrices for each weight matrix in the Transformer layers. At each iteration $t$, the algorithm performs the following steps:

i. **Learn LoRA Adaptations**: For each weight matrix $\boldsymbol{W}_l$ in layer $l$ of a Transformer, learn the corresponding LoRA matrices $\boldsymbol{A}_l^t$ and $\boldsymbol{B}_l^t$ by minimizing the loss function $\mathcal{L}_t$ defined over the current model predictions and the target outputs:

$$\mathcal{L}_t = \sum_{i=1}^N \ell\big(\mathcal{M}_{t-1}(\boldsymbol{x}_i), y_i\big) + \lambda \sum_{l=1}^L \big(\|\boldsymbol{A}_l^t\|_F^2 + \|\boldsymbol{B}_l^t\|_F^2\big), \tag{7}$$

where $\mathcal{M}_{t-1}(\boldsymbol{x}_i)$ denotes the output of the model from the previous iteration for the input $\boldsymbol{x}_i$, $\ell(\cdot)$ is a differentiable loss function (*e.g.*, cross-entropy), $\lambda$ is a regularization coefficient, and $L$ is the number of Transformer layers.

ii. **Merge LoRA Adaptations**: Merge the learned LoRA matrices into the corresponding weight matrices of the Transformer model to obtain the updated model $\mathcal{M}_t$:

$$\boldsymbol{W}_l^t \leftarrow \boldsymbol{W}_l^{t-1} + \alpha_t(\boldsymbol{A}_l^t \boldsymbol{B}_l^t), \quad \text{for} \quad l = 1, \ldots, L, \tag{8}$$

where $\boldsymbol{W}_l^t$ represents the adapted weight matrix of layer $l$ at iteration $t$, $\boldsymbol{W}_l^{t-1}$ is the weight matrix from the previous iteration, and $\alpha_t$ is a learning rate that controls the contribution of the LoRA. By absorbing the learning rate $\alpha_t$ into the LoRA matrices, we simplify the notation and make it clear that the effective update to the weight matrices is directly determined by the learned LoRAs. Thus in our case $\alpha_t = 1$.

Figure 3 shows the overall pipeline of XGBLoRA. The algorithm iterates for a fixed number of iterations $T$ or until a convergence criterion is met. At each iteration $t$, a new set of LoRA matrices is learned based on the current state of the Transformer model, and then merged into the corresponding weight matrices to update the model parameters. This iterative process allows for progressive refinement of the model's predictions by repeatedly learning and integrating LoRAs.

**GB Iterations and Training Steps.** Let $K$ be the total number of training step which is usually fixed for the fine-tuning. The typical GB set the the number of iteration $T$ to the number of training steps $K$. In this case, XGBLoRA booster is trained for only one step (one backpropagation) during each GB iteration and merge to the base model. However, to maintain the minimal prediction ability of the booster, each booster is trained for $\kappa$ steps within one GB iterations (Figure 3). As a result, the relation between total number of training step $K$ and GB iterations $T$ is described as $K = \kappa T$.

**Relationship to LoRA.** It is worth noting that the original LoRA method (Hu et al., 2021) can be regarded as a variant of XGBLoRA with one iteration ($T = 1$) where $\kappa = K$. In this case, only one LoRA booster is merged with the base model $\mathcal{M}_0$ at the end of training.

### 3.3 Understanding Ensemble of Weak Learners.

The crucial principle of Gradient Boosting is building a strong ensemble model with *weak learners*. For instance, a very successful gradient boosting method, *i.e.*, Gradient Boost Decision Tree (GBDT) (Chen & Guestrin, 2016) artificially limits the tree boosters to a very shallow depth (usually only 1 split) to ensure that each booster is only slightly better than the random decision. Thus, boosting algorithms are highly resilient against noisy data and overfitting (Friedman, 2002). Since the individual booster is too simple to overfit, it is very hard to combine them in a way that the strong ensemble would overfit to the whole training data. Adhering to this principle, below we design the following strategies for controlling the the expressiveness of each LoRA boosters.

Table 1: Comparison of computational costs. By definition, LoRA uses rank $r \triangleq R$ and updates all layers $l \triangleq L$ at once. XGBLoRA uses $r \ll R$ and $l \ll L$. Moreover, $\alpha$: the comp. cost for a LoRA ($r \triangleq R$) adapter in one transformation layer; $\beta$: the comp. cost for a base model; $L$: the total number of layers; $K = \kappa T$: total training steps; $T$: no. of Gradient Boosters (alg. iterations).

| | Cost/learner $l\alpha r/R$ | Step/iteration $\kappa$ | #iterations $T$ | Total Cost $l\alpha\kappa T + \beta$ |
|---|---|---|---|---|
| LoRA ($r \triangleq R, l \triangleq L$) | $L\alpha$ | $K$ | 1 | $L\alpha K + \beta$ (Upper Bound) |
| XGBLoRA ($r = R, l = L/3$) | $L\alpha/3$ | $K/10$ | 10 | $L\alpha K/3 + \beta$ |
| XGBLoRA ($r = 1, l = L/3$) | $L\alpha/3R$ | $K/10$ | 10 | $L\alpha K/3R + \beta$ |

**The Rank-1 update.** The rank $r$ of the LoRA matrices is a hyperparameter that controls the expressiveness and efficiency of the adaptation. A smaller rank leads to more parameter-efficient adaptations but may limit the ability to capture complex patterns in the residual errors. On the other hand, a larger rank allows for more expressive adaptations but increases the computational and memory requirements. In light of the gradient boosting theory, rank-1 updates provides the right balance between approximation power and regularization. They allow for a more fine-grained, step-by-step approximation of the target function due to a larger number of weaker learners in ensemble, leading to better generalization.

**Random Layer Selection.** This strategy resembles the random column selection strategy in GBDT, which limits complexity but increases diversity among boosters. Instead of modifying all layers of the language model (LM), we randomly select $L_s$ ($L_s \leq L$) layers to add LoRAs for building the booster. By adapting only a subset of layers in each iteration, the booster's ability to make drastic changes is limited. This intentional constraint ensures that each booster remains 'weak' in its predictive power (core principle of GB). Moreover, this strategy injects randomness into the final ensemble model, creating diversity among the boosters. Each booster focuses on different parts of the model, capturing distinct aspects of the data. This diversity is crucial for the success of ensemble methods.

> XGBLoRA offers several advantages over other fine-tuning approaches and the standard LoRA.
> 1) **Ensemble Learning**: By iteratively learning and merging LoRAs, XGBLoRA can efficiently adapt the pre-trained model to the downstream task. The iterative nature of the algorithm allows for progressive refinement of model predictions, leading to better generalization performance without explicitly adding regularization terms to the loss function.
> 2) **The Weak Learner Principle:** By rank-1 updates and random layer adaptation XGBLoRA significantly reduces computational cost, achieving efficient LLM fine-tuning. XGBLoRA enables rapid, incremental adaptation incurring low memory overhead, making it particularly suitable for fine-tuning LLMs where full parameter updates are prohibitively costly.

## 3.4 COMPUTATIONAL COSTS

Table 1 compares the cost of XGBLoRA and LoRA. XGBLoRA's computational cost is upper-bounded by the LoRA's cost, as XGBLoRA selects fewer LoRA layers (random selection) and uses lower ranks for training (rank-1 updates). The computational costs incurred by these two approaches are equal only when XGBLoRA selects *ALL* layers and uses the *same* rank as LoRA.

## 3.5 THEORETICAL ANALYSIS OF GRADIENT BOOSTING LoRA

In this subsection, we present a theoretical analysis of the eXtreme Gradient Boosting LoRA (XGBLoRA) framework for Transformer-based language models. Our analysis aims to provide convergence guarantees and approximation error bounds for the proposed method. We begin by introducing necessary definitions and then present key lemmas and theorems.

**Notation.** Consider a neural network with $L$ layers: $f(\mathbf{x}) = f_L \circ f_{L-1} \circ \cdots \circ f_2 \circ f_1(\mathbf{x})$, where $f_1(\mathbf{x}) = \mathbf{W}_1\mathbf{x}$ is an embedding layer. Moreover, $f_i(\mathbf{x}) = \sigma(\mathbf{W}_i\mathbf{x})$ for $i = 2, \ldots, L - 1$, where $\sigma$ is a Lipschitz continuous activation function. $f_L(\mathbf{x}) = \phi(\mathbf{W}_L\mathbf{x})$, where $\phi$ is a convex function. $\mathbf{W}_i \in \mathbb{R}^{d_i \times d_{i-1}}$ are weight matrices.

We present three key lemmas that are crucial for our convergence and expressiveness theorems:

**Lemma 1 (**XGBLoRA **Gradient Approximation.)** *The* XGBLoRA *update approximates the full gradient update with error:*

$$\left\|\nabla_{\mathbf{W}^{(t)}+\mathbf{A}^{(t)}\mathbf{B}^{(t)T}}\mathcal{L}(\mathbf{W}^{(t)}+\mathbf{A}^{(t)}\mathbf{B}^{(t)T})-\mathbf{A}^{(t)}\mathbf{B}^{(t)T}\right\|_F \leq \frac{C_1}{\sqrt{r}}+\frac{C_2}{\sqrt{M}}, \tag{9}$$

*where $r$ is the LoRA rank, $M$ is the number of minibatches, and $C_1, C_2$ are constants depending on the properties of $\mathcal{L}$ and the gradient variance, respectively. (The complete proof is in the Appendix.)*

**Lemma 2 (Accumulated Update Bound.)** *For the* XGBLoRA *update process:*

$$\|\mathbf{A}^{(t)}\|_F \leq \eta_m M G \quad and \quad \|\mathbf{B}^{(t)}\|_F \leq \eta_m M G, \tag{10}$$

*where $G$ is an upper bound on $\|\nabla_{\mathbf{W}^{(t)}+\mathbf{A}^{(t)}\mathbf{B}^{(t)T}}\mathcal{L}(\mathbf{W}^{(t)}+\mathbf{A}^{(t)}\mathbf{B}^{(t)T})\|_F$. (The complete proof is in the Appendix.)*

**Lemma 3 (Gradient Lipschitz Continuity.)** *For any two weight matrices $\mathbf{W}_1$ and $\mathbf{W}_2$:*

$$\|\nabla_{\mathbf{W}_1}\mathcal{L}(\mathbf{W}_1)-\nabla_{\mathbf{W}_2}\mathcal{L}(\mathbf{W}_2)\|_F \leq L'\|\mathbf{W}_1-\mathbf{W}_2\|_F, \tag{11}$$

*where $L'$ is the Lipschitz constant of the gradient. (The complete proof can be found in the Appendix.)*

We now present our main convergence theorem for XGBLoRA:

**Theorem 1 (**XGBLoRA **Convergence.)** *Under the* XGBLoRA *update process, assuming $\beta$-smoothness and $\mu$-strong convexity of $\mathcal{L}$, after $T$ iterations:*

$$\mathbb{E}\big[\mathcal{L}(\mathbf{W}^{(T)})\big] - \mathcal{L}^* \leq \frac{C_3}{\sqrt{T}}+\frac{C_4}{M\sqrt{T}}+\epsilon(r), \tag{12}$$

*where $C_3$ and $C_4$ are constants depending on $\beta, \mu, G, \eta_m, L'$, and $\epsilon(r) = \frac{C_5}{r}$ for some constant $C_5$. $N$ is the number of samples. (The complete proof is in the Appendix.)*

> **Remark 1** *The error term $\epsilon(r) = \frac{C_5}{r}$ suggests that while higher ranks can reduce this error, the benefit diminishes as $r$ increases. However, the theorem implies that increasing the number of iterations $T$ can compensate for a lower rank $r$. By using rank-1 updates and increasing $T$, we can achieve a similar convergence rate to higher-rank methods while maintaining lower computational complexity per iteration. It supports that* XGBLoRA *using multiple simple (rank-1) weak learners in an ensemble, rather than fewer complex (higher-rank) learners, to efficiently approximate the target function.*

Below we also present a theorem characterizing the expressive power of XGBLoRA:

**Theorem 2 (**XGBLoRA **Expressiveness Error.)** *Let $f^*$ be any function in the original function class, and $f_T$ be the function represented by the* XGBLoRA*-updated network after $T$ iterations. Then:*

$$\mathbb{E}_{\mathbf{x}\sim\mathcal{D}}\big[(f_T(\mathbf{x})-f^*(\mathbf{x}))^2\big] \leq C_6\Big(\frac{1}{r}+\frac{1}{M\sqrt{M}T}+\frac{1}{\sqrt{T}}\Big), \tag{13}$$

*where $C_6$ is a constant depending on the network architecture, the Lipschitz constants of the activation functions, and $L'$. (The complete proof can be found in the Appendix.)*

> **Remark 2** *The theorem shows that the approximation error can be reduced by either increasing $r$ or $T$. Let $\epsilon^* = \mathbb{E}_{\mathbf{x}\sim\mathcal{D}}[(f_T(\mathbf{x})-f^*(\mathbf{x}))^2]$ be the desired approximation that achieves the optimal generalization. Theorem 2 explicitly reveals that to maintain $\epsilon^*$, one can trade off LoRA rank $r$ against iterations $T$. Original LoRA is the case where $T = 1$ and $r \gg 1$. In contrast,* XGBLoRA *enjoys the opposite setting $r = 1$ and $T \gg 1$. Thus,* XGBLoRA *can maintain high expressiveness and generalization capability while keeping each individual update computationally efficient. Note that increase $T$ does not mean more total training steps $K$. As $K$ is usually fixed, $T$ can be adjusted via booster's training steps $\kappa$ where $T = \frac{K}{\kappa}$ (discussion see section 4.1).*

Theorems 1 & 2 justify the use of rank-1 updates and explain the effectiveness of XGBLoRA achieved by leveraging ensemble learning and iterative refinement while maintaining low memory footprint.

Table 2: Results on GLUE for natural language understanding tasks. We report the overall (matched and mismatched) accuracy for MNLI, Matthew's correlation for CoLA, Pearson correlation for STS-B, and accuracy for other tasks. Higher is better for all metrics. We also report the number of trainable parameters (#Params) for each method. * indicates results extracted from (Ren et al., 2024)

| Method | #Params | MNLI | SST-2 | CoLA | QQP | QNLI | RTE | MRPC | STS-B | Avg |
|---|---|---|---|---|---|---|---|---|---|---|
| Fully FT | 1000‰ | 87.62 | 94.84 | 64.58 | 91.87 | 92.80 | 70.80 | 90.20 | 91.23 | 86.87 |
| BitFit | 0.82‰ | 85.29 | 94.61 | 59.58 | 88.10 | 91.20 | 78.80 | 88.73 | 90.32 | 84.70 |
| HAdapter | 2.50‰ | 87.45 | 94.72 | 63.88 | 90.29 | 92.71 | 83.39 | 89.22 | 90.80 | 86.15 |
| PAdapter | 2.43‰ | 87.11 | 94.15 | 62.74 | 89.95 | 92.71 | 84.48 | 87.99 | 90.13 | 85.62 |
| DyLoRA* | 2.65‰ | 86.33 | 94.26 | 61.12 | 90.17 | 92.22 | 84.47 | 89.46 | 91.06 | 86.14 |
| DeltaLoRA* | 2.65‰ | 87.50 | 95.06 | 63.82 | 90.87 | 93.09 | 87.00 | 90.19 | 91.57 | 87.38 |
| MELoRA* | 2.65‰ | 87.20 | 95.41 | 64.09 | 90.77 | 93.11 | 86.64 | **90.93** | **91.93** | 87.52 |
| LoRA | 2.65‰ | 87.20 | 94.38 | 65.61 | 89.25 | 92.07 | 85.59 | 87.99 | 91.01 | 86.63 |
| TriLoRA | 2.65‰ | 86.81 | 94.61 | 64.47 | 89.61 | 91.82 | 76.53 | 88.24 | 90.31 | 85.30 |
| AdaLoRA | 2.65‰ | 87.31 | 94.72 | 64.33 | 89.77 | 92.81 | 85.95 | 88.24 | 90.48 | 86.70 |
| FLoRA | 2.65‰ | 87.31 | 94.38 | 64.09 | 89.97 | 92.77 | 85.67 | 87.75 | 90.77 | 86.86 |
| DoRA | 3.32‰ | 86.74 | 94.50 | 66.19 | 90.28 | 91.95 | 85.78 | 88.48 | 91.01 | 87.11 |
| ReLoRA | 21.2‰ | **89.06** | 95.38 | 64.72 | 90.74 | **93.68** | 84.72 | 89.65 | 90.53 | 87.30 |
| XGBLoRA | **0.21‰** | 87.91 | **95.70** | **66.28** | **91.04** | 93.36 | **86.10** | 90.57 | 91.84 | **87.85** |

Table 3: Accuracy comparison of LLaMA 7B/13B, LLaMA2 7B, and LLaMA3 8B with various PEFT methods on eight commonsense reasoning datasets. Results of baseline methods (*) on LLaMA 7B/13B are extracted from (Hu et al., 2023). Results of LoRA on LLaMA2 7B and LLaMA3 8B are obtained using the hyperparameters described in Hu et al. (2023)

| | Method | #Params | BoolQ | PIQA | SIQA | HellaS. | WinoG. | ARC-e | ARC-c | OBQA | Avg |
|---|---|---|---|---|---|---|---|---|---|---|---|
| LLaMA-7B | Prefix* | 1.10‰ | 64.3 | 76.8 | 73.9 | 42.1 | 72.1 | 72.9 | 54.0 | 60.6 | 64.6 |
| | Series* | 9.90‰ | 63.0 | 79.2 | 76.3 | 67.9 | 75.7 | 74.5 | 57.1 | 72.4 | 70.8 |
| | Parallel* | 35.4‰ | 67.9 | 76.4 | 78.8 | 69.8 | 78.9 | 73.7 | 57.3 | 75.2 | 72.3 |
| | LoRA* | 8.30‰ | 68.9 | 80.7 | 77.4 | 78.1 | 78.8 | 77.8 | 61.3 | 74.8 | 74.7 |
| | XGBLoRA | 0.34‰ | **69.1** | **82.6** | 77.3 | **86.1** | 80.2 | 80.5 | 65.3 | 78.5 | **77.40** |
| LLaMA-13B | Prefix* | 0.30‰ | 65.3 | 75.4 | 72.1 | 55.2 | 68.6 | 79.5 | 62.9 | 68.0 | 68.4 |
| | Series* | 8.00‰ | 71.8 | 83.0 | 79.2 | 88.1 | 82.4 | 82.5 | 67.3 | 81.8 | 79.5 |
| | Parallel* | 28.0‰ | 72.5 | 84.8 | 79.8 | 92.1 | 84.7 | 84.2 | 71.2 | 82.4 | 81.5 |
| | LoRA* | 6.70‰ | 72.1 | 83.5 | 80.5 | 90.5 | 83.7 | 82.8 | 68.3 | 82.4 | 80.5 |
| | XGBLoRA | 0.27‰ | **72.7** | **85.1** | 81.8 | **92.7** | 84.5 | 84.5 | 69.9 | 83.1 | **81.8** |
| LLaMA3-8B | LoRA | 7.00‰ | 72.1 | 83.5 | **80.5** | 90.5 | 83.7 | 82.8 | 68.3 | 82.4 | 80.5 |
| | XGBLoRA | 0.30‰ | **72.5** | **85.8** | 78.3 | **93.5** | 83.9 | 88.1 | 75.1 | 84.2 | **83.0** |

## 4 EXPERIMENTAL EVALUATION

**Experiment Settings.** We set the rank of XGBLoRA to $r = 1$ and rank of LoRA to $r = 8$ as default. The number of sampled layers for XGBLoRA is $L_s = 8$. To ensure a fair comparison, we initially fine-tuned models with XGBLoRA following the LoRA configuration, *e.g.*, weight initialization, learning rate, *etc*. (Hu et al., 2021), and maintained the same training steps $K$ for both XGBLoRA and LoRA when fine-tuning on the same datasets. Since $K$ is fixed, the number of iterations $T$ for gradient boosting is calculated as $T = \frac{K}{\kappa}$. The training steps for each booster is set to $\kappa = 8$ to maintain minimal prediction power. We conduct experiment on three tasks including the GLUE benchmark, commonsense reasoning, and MMLU. The codebases for baselines implementation and evaluation are sourced from their official GitHub repositories/library (*i.e.*, Commonsense Reasoning, GLUE, and MMLU are from Hu et al. (2023), Si et al. (2024), Zheng et al. (2024), respectively).

**GLUE Benchmark.** In GLUE experiments, we employed one small scales of transformer, *RoBERTa-base* (Liu, 2019), as the base model. We used the General Language Understanding Evaluation (GLUE) (Raffel et al., 2020a) benchmark as our dataset, which comprises two single-sentence classification tasks, three similarity and paraphrase tasks, and four natural language inference tasks. Details of the GLUE dataset are provided in Table 7 (Appendix). There are two prominent series of extension-based methods within parameter-efficient tuning. The first series, the Adapter derivatives, comprises methods such as those introduced by Houlsby et al. (2019a), Houlsby et al. (2019a), and introduced by Pfeiffer et al. (2020); Zaken et al. (2021), which incorporate small-scale neural modules, or adapters, into existing architectures. The second series, known as LoRA derivatives, includes developments such as LoRA (Hu et al., 2021), AdaLoRA (Zhang et al., 2023), TriLoRA (Feng et al., 2024), FLoRA (Hao et al., 2024), DoRA (Liu et al., 2024), and DyLoRA (Valipour et al., 2023), AdaLoRA (Zhang et al., 2023), Delta-LoRA (Zi et al., 2023), MeLoRA (Ren et al., 2024), and ReLoRA (Lialin et al., 2023).

Table 4: MMLU scores for `XGBLoRA` and other PEFT methods, showcasing `XGBLoRA`'s ability to achieve high performance while maintaining parameter efficiency across base models. Best performance is indicated by the bold face numbers.

| | FT-Method | # Params | STEM | Social | Human | Other | Average |
|---|---|---|---|---|---|---|---|
| LLaMA3-8B | FT | 1000‰ | 52.93 | 73.40 | 59.06 | 69.34 | 63.26 |
| | LoRA | 7.00‰ | 54.45 | 74.82 | 58.96 | 70.23 | 64.10 |
| | XGBLoRA | 0.30‰ | **54.93** | **75.40** | **61.06** | **71.34** | **65.37** |
| Mistral-7B | FT | 1000‰ | 50.00 | 68.07 | 53.12 | 65.01 | 58.09 |
| | LoRA | 8.30‰ | **50.60** | 68.87 | 53.62 | 65.21 | 58.99 |
| | XGBLoRA | 0.34‰ | 50.40 | **69.04** | **54.28** | **65.46** | **59.26** |
| LLaMA2-13B | FT | 1000‰ | 46.23 | 64.47 | 49.34 | 61.23 | 55.31 |
| | LoRA | 6.70‰ | 46.56 | 64.77 | 49.67 | 61.76 | 55.69 |
| | XGBLoRA | 0.27‰ | **46.70** | **65.64** | **50.56** | **62.46** | **56.34** |

Table 5: Performance comparison of `XGBLoRA` in LlaMA3-8B with different rank values ($r$) and other fine-tuning methods on the MMLU and Commonsense Reasoning benchmark.

(a) MMLU benchmark.

| Method | #Params. | STEM | Social. | Human. | Other | **Average** |
|---|---|---|---|---|---|---|
| FT | 1000‰ | 54.35 | 74.62 | 58.86 | 70.03 | 63.82 |
| LoRA ($r = 8$) | 7.00‰ | 54.45 | 74.82 | 58.96 | 70.23 | 64.10 |
| XGBLoRA ($r = 16$) | 4.80‰ | 55.00 | 75.30 | 60.68 | 70.94 | 65.03 |
| XGBLoRA ($r = 8$) | 2.40‰ | 54.93 | 75.33 | 61.06 | 71.25 | 65.23 |
| XGBLoRA ($r = 4$) | 1.20‰ | 55.10 | **75.56** | 60.98 | 71.44 | 65.33 |
| XGBLoRA ($r = 1$) | 0.30‰ | **55.20** | 75.43 | **61.19** | **71.31** | **65.36** |

(b) Commonsense Reasoning benchmark.

| Method | #Params | BoolQ | PIQA | SIQA | HellaS. | WinoG. | ARC-e | ARC-c | OBQA | Avg |
|---|---|---|---|---|---|---|---|---|---|---|
| LoRA | 7.00‰ | 70.8 | 85.2 | 79.9 | 91.7 | **84.3** | **84.2** | 71.2 | 79.0 | 80.8 |
| XGBLoRA ($r = 16$) | 4.80‰ | **72.5** | 84.3 | **80.9** | 90.1 | 82.9 | 82.7 | 69.7 | 83.6 | 80.8 |
| XGBLoRA ($r = 4$) | 1.20‰ | 72.4 | 84.9 | 81.5 | 92.4 | 84.2 | **84.2** | 69.6 | 82.8 | 81.5 |
| XGBLoRA ($r = 1$) | 0.30‰ | **72.5** | 85.8 | 78.3 | **93.5** | 83.7 | **88.1** | **75.1** | 84.2 | **83.0** |

**Commonsense Reasoning.** The commonsense reasoning tasks comprise 8 sub-tasks, each with a predefined training and testing set. We follow the setting of Hu et al. (2023) and amalgamate the training datasets from all 8 tasks to create the final training dataset and conduct evaluations on the individual testing dataset for each task. We evaluate `XGBLoRA` against LoRA and baselines: Prompt learning (Prefix) (Li & Liang, 2021), Series adapter (Series) (Houlsby et al., 2019b), and Parallel adapter (Parallel) (He et al., 2021) on *LLaMA7B/13B* and *LLaMA3-8B* (Touvron et al., 2023a).

**MMLU.** We evaluate the downstream task performance of `XGBLoRA` on 3 language models *LLaMA3-8B* (inc, 2024), *Mistral-7B* (Jiang et al., 2023), and *LLaMA2-13B* (Touvron et al., 2023a). We employ the instruction-following finetuning task with Alpaca GPT-4(en) dataset, which consists instances generated by GPT-4 (OpenAI, 2023) based on inputs from Alpaca (Taori et al., 2023). We adopt the **The Massive Multitask Language Understanding benchmark (MMLU)** (Hendrycks et al., 2020) to test our model. It consists of multiple-choice questions in humanities, social sciences, and STEM.

Tables 2, 3 & 4 compare various fine-tuning methods, including full fine-tuning (Fully FT), different LoRA variants, and the proposed `XGBLoRA` method. `XGBLoRA` demonstrates strong performance across various tasks. It consistently achieves the highest or near-highest scores among the PEFT methods across all base models and subject domains. This suggests that `XGBLoRA` is generally an effective approach for different base models, making it a robust choice for PEFT. It is worth noting that benefiting from the principle of weak learners, `XGBLoRA` achieves strong performance with significantly fewer parameters than other methods, including standard LoRA. These findings strongly support our claims about `XGBLoRA`'s ability to bridge the performance gap and maintain efficiency.

## 4.1 Investigating the Weak Learner of Gradient Boosting

**Number of Weak Learners (Iteration).** The number of weak learners in `XGBLoRA` is equal to the number of the iterations in gradient boosting. Since the total train step $K$ is fixed per dataset. the number of iteration $T$ is controlled by the merged/training interval $\kappa$ for each booster. Thus, $T = \frac{K}{\kappa}$. Large/small $\kappa$ indicates fewer/more weak learners (iteration). The results of varying $\kappa$ are presented in Figure 4: having more weak learners in the Gradient Boosting LoRA (`XGBLoRA`) framework leads to better performance. This reinforces the following points for `XGBLoRA`:

i. **Iterative refinement:** With more weak learners, `XGBLoRA` can perform more iterations of refinement, allowing the model to progressively improve its predictions and capture more complex

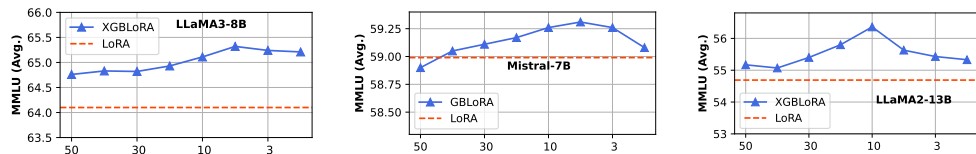

Figure 4: Performance of XGBLoRA with varying $\kappa = \frac{K}{T}$ for LLaMA3-8B, Mistral-7B, and LLaMA2-13B base models. Perfomance of LoRA is marked as red dash line

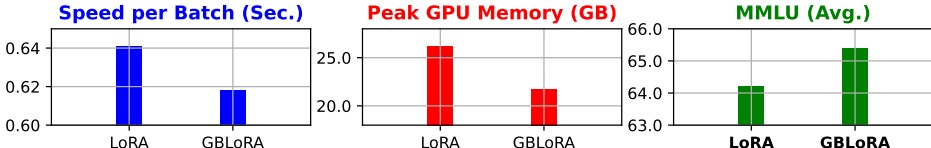

Figure 5: Memory and Computation Efficient Fine-tuning with Small Weak Learners (LLaMA3-8B).

Table 6: Impact of the number of adapted layers ($L_s$) on the performance of XGBLoRA($r = 8$) in LlaMA3-8B compared to full fine-tuning (FT) and LoRA ($r = 8$) on the MMLU benchmark.

| FT-Method | # Param. | STEM | Social. | Human. | Other | Average |
|---|---|---|---|---|---|---|
| LoRA($r = 8$) | 7.00‰ | 54.45 | 74.82 | 58.96 | 70.23 | 64.10 |
| XGBLoRA ($L_s = 2$) | 0.42‰ | 54.80 | 75.13 | 61.07 | 70.87 | 65.07 |
| XGBLoRA ($L_s = 4$) | 0.84‰ | 54.90 | 75.50 | 61.04 | 71.07 | 65.21 |
| XGBLoRA ($L_s = 11$) | 2.33‰ | **54.93** | **75.50** | **61.32** | **71.41** | **65.38** |
| XGBLoRA ($L_s = 16$) | 3.50‰ | 55.13 | 75.56 | 61.02 | 71.53 | 65.37 |
| XGBLoRA ($L_s = 33$) | 7.00‰ | 55.13 | 75.53 | 61.04 | 71.38 | 65.33 |

patterns in the data. Each additional weak learner focuses on the residual errors from the previous iterations, enabling the model to make finer-grained adjustments.

ii. **Ensemble effect:** As XGBLoRA combines multiple weak learners learned across different iterations, having more weak learners leads to a more diverse and robust ensemble. This helps reduce the bias and improves the overall performance of the adapted model.

Note that with a large number of weak learners, there is a risk of performance degradation. As the number of iterations $T$ increases, the training interval $\kappa$ for each booster decreases. This aligns with the principle of weak learners in traditional GB methods. While we want each booster to be 'weak' to prevent overfitting, they have to possess enough predictive power to contribute to the ensemble.

**Complexity of Weak Learners.** Table 5 shows the role of rank $r$ in XGBLoRA's weak learners. XGBLoRA with smaller ranks outperforms LoRA ($r = 8$) and XGBLoRA with larger ranks ($r = 16$), corroborating our discussion on weak learner complexity. The strong performance of XGBLoRA with low-rank adaptations suggests that combining multiple simple weak learners can effectively capture complex patterns and improve generalization. This highlights the ensemble effect in XGBLoRA, leading to strong performance while maintaining parameter efficiency.

Table 6 further showcases the effect of random layer selection. XGBLoRA outperforms full fine-tuning (FT) and LoRA ($r = 8$), while adapting parts of the layers. With just 4 adapted layers ($L_s = 4$), XGBLoRA surpasses LoRA and performs comparably to FT using significantly fewer trainable parameters (2.3‰). This demonstrates its ability to leverage gradient boosting and the ensemble effect of weak learners to achieve a strong performance with minimal computational overhead.

**Memory and Computational Efficiency with Weak Learners.** Figure 5 illustrates the superior performance of XGBLoRA compared to standard LoRA in terms of the MMLU average score, while simultaneously consuming less memory and requiring less time per batch. This empirical evidence not only underscores the advantages of XGBLoRA, but also suggests its potential for scaling to fine-tune larger language models for which GPU memory constraints are often significant bottlenecks.

## 5 CONCLUSIONS

We have proposed XGBLoRA for fine-tuning LLMs in a parameter-efficient manner by posing fine-tuning as a gradient boosting where LoRA matrices are used as weak learners to be iteratively combined to form a strong ensemble model. We provide theoretical analysis establishing the convergence and approximation error of XGBLoRA, highlighting the interplay between the LoRA rank, expressiveness, and the number of boosting iterations. Extensive experiments demonstrate the effectiveness of XGBLoRA, which consistently outperforms the standard LoRA while maintaining parameter/computational efficiency. Broader Impact & Limitations are in Appendices D & E.

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

## A    DETAILED PROOFS FOR XGBLORA LEMMAS

**Lemma 4 (XGBLORA Gradient Approximation)** *The XGBLoRA update approximates the full gradient update with error:*

$$\|\nabla_{\mathbf{W}^{(t)}+\mathbf{A}^{(t)}\mathbf{B}^{(t)T}}\mathcal{L}(\mathbf{W}^{(t)}+\mathbf{A}^{(t)}\mathbf{B}^{(t)T})-\mathbf{A}^{(t)}\mathbf{B}^{(t)T}\|_F \leq \frac{C_1}{\sqrt{r}}+\frac{C_2}{\sqrt{M}}$$

*where $r$ is the LoRA rank, $M$ is the number of minibatches, and $C_1, C_2$ are constants.*

**Proof 1** *1) Let $\mathbf{G} = \nabla_{\mathbf{W}^{(t)}+\mathbf{A}^{(t)}\mathbf{B}^{(t)T}}\mathcal{L}(\mathbf{W}^{(t)}+\mathbf{A}^{(t)}\mathbf{B}^{(t)T})$ be the true gradient.*

*2) The XGBLoRA update $\mathbf{A}^{(t)}\mathbf{B}^{(t)T}$ can be seen as an approximation of $\mathbf{G}$.*

*3) Let $\mathbf{G}_r$ be the best rank-r approximation of $\mathbf{G}$. By the Eckart-Young-Mirsky theorem:*

$$\|\mathbf{G}-\mathbf{G}_r\|_F \leq \frac{\|\mathbf{G}\|_*}{\sqrt{r}} \leq \frac{C_1}{\sqrt{r}}$$

*where $\|\cdot\|_*$ is the nuclear norm and $C_1$ is a constant depending on the properties of $\mathcal{L}$.*

*4) The XGBLoRA update $\mathbf{A}^{(t)}\mathbf{B}^{(t)T}$ is computed using M minibatches. Let $\mathbf{G}_j$ be the gradient estimate from the j-th minibatch. Then:*

$$\mathbf{A}^{(t)}\mathbf{B}^{(t)T} \approx \frac{1}{M}\sum_{j=1}^{M}\mathbf{G}_j$$

*5) By the law of large numbers and assuming bounded variance of gradient estimates:*

$$\|\frac{1}{M}\sum_{j=1}^{M}\mathbf{G}_j - \mathbf{G}\|_F \leq \frac{C_2}{\sqrt{M}}$$

*where $C_2$ is a constant related to the gradient variance.*

*6) Combining these bounds using the triangle inequality:*

$$\|\mathbf{G}-\mathbf{A}^{(t)}\mathbf{B}^{(t)T}\|_F \leq \|\mathbf{G}-\mathbf{G}_r\|_F + \|\mathbf{G}_r-\mathbf{A}^{(t)}\mathbf{B}^{(t)T}\|_F \leq \frac{C_1}{\sqrt{r}}+\frac{C_2}{\sqrt{M}}$$

*This completes the proof.*

**Lemma 5 (Accumulated Update Bound)** *For the XGBLoRA update process:*

$$\|\mathbf{A}^{(t)}\|_F \leq \eta_m MG \quad and \quad \|\mathbf{B}^{(t)}\|_F \leq \eta_m MG$$

*where $G$ is an upper bound on $\|\nabla_{\mathbf{W}^{(t)}+\mathbf{A}^{(t)}\mathbf{B}^{(t)T}}\mathcal{L}(\mathbf{W}^{(t)}+\mathbf{A}^{(t)}\mathbf{B}^{(t)T})\|_F$.*

**Proof 2** *1) Recall the update rule for $\mathbf{A}^{(t)}$:*

$$\mathbf{A}^{(t)} \leftarrow \mathbf{A}^{(t)} - \eta_m \nabla_{\mathbf{W}^{(t)}+\mathbf{A}^{(t)}\mathbf{B}^{(t)T}}\mathcal{L}(\mathbf{W}^{(t)}+\mathbf{A}^{(t)}\mathbf{B}^{(t)T})\mathbf{B}^{(t)}$$

*2) Taking the Frobenius norm and applying the triangle inequality:*

$$\|\mathbf{A}^{(t)}\|_F \leq \|\mathbf{A}^{(t-1)}\|_F + \eta_m\|\nabla_{\mathbf{W}^{(t)}+\mathbf{A}^{(t)}\mathbf{B}^{(t)T}}\mathcal{L}(\mathbf{W}^{(t)}+\mathbf{A}^{(t)}\mathbf{B}^{(t)T})\|_F\|\mathbf{B}^{(t)}\|_F$$

*3) Using the gradient bound $\|\nabla_{\mathbf{W}^{(t)}+\mathbf{A}^{(t)}\mathbf{B}^{(t)T}}\mathcal{L}(\mathbf{W}^{(t)}+\mathbf{A}^{(t)}\mathbf{B}^{(t)T})\|_F \leq G$:*

$$\|\mathbf{A}^{(t)}\|_F \leq \|\mathbf{A}^{(t-1)}\|_F + \eta_m G\|\mathbf{B}^{(t)}\|_F$$

*4) Applying this inequality recursively for all M minibatches, and noting that $\mathbf{A}^{(t)}$ is initialized to $\mathbf{0}$:*

$$\|\mathbf{A}^{(t)}\|_F \leq \eta_m MG\|\mathbf{B}^{(t)}\|_F$$

*5) Similarly for $\mathbf{B}^{(t)}$, we can derive:*

$$\|\mathbf{B}^{(t)}\|_F \leq \eta_m MG \|\mathbf{A}^{(t)}\|_F$$

*6) Combining these inequalities:*

$$\|\mathbf{A}^{(t)}\|_F \leq \eta_m MG \quad and \quad \|\mathbf{B}^{(t)}\|_F \leq \eta_m MG$$

*This completes the proof.*

**Lemma 6 (Gradient Lipschitz Continuity)** *For any two weight matrices $\mathbf{W}_1$ and $\mathbf{W}_2$:*

$$\|\nabla_{\mathbf{W}_1}\mathcal{L}(\mathbf{W}_1) - \nabla_{\mathbf{W}_2}\mathcal{L}(\mathbf{W}_2)\|_F \leq L'\|\mathbf{W}_1 - \mathbf{W}_2\|_F$$

*where L is the Lipschitz constant of the gradient.*

**Proof 3** *1) This lemma is a standard assumption in optimization theory, often referred to as the smoothness condition.*

*2) It can be derived from the assumption that the Hessian of $\mathcal{L}$ is bounded:*

$$\|\nabla^2\mathcal{L}(\mathbf{W})\|_2 \leq L \quad \forall \mathbf{W}$$

*where $\|\cdot\|_2$ denotes the spectral norm.*

*3) By the mean value theorem, there exists a $\mathbf{W}_t = t\mathbf{W}_1 + (1-t)\mathbf{W}_2$ for some $t \in [0,1]$ such that:*

$$\nabla_{\mathbf{W}_1}\mathcal{L}(\mathbf{W}_1) - \nabla_{\mathbf{W}_2}\mathcal{L}(\mathbf{W}_2) = \nabla^2\mathcal{L}(\mathbf{W}_t)(\mathbf{W}_1 - \mathbf{W}_2)$$

*4) Taking the Frobenius norm of both sides:*

$$\|\nabla_{\mathbf{W}_1}\mathcal{L}(\mathbf{W}_1) - \nabla_{\mathbf{W}_2}\mathcal{L}(\mathbf{W}_2)\|_F = \|\nabla^2\mathcal{L}(\mathbf{W}_t)(\mathbf{W}_1 - \mathbf{W}_2)\|_F$$

*5) Using the property that $\|\mathbf{A}\mathbf{B}\|_F \leq \|\mathbf{A}\|_2\|\mathbf{B}\|_F$:*

$$\|\nabla^2\mathcal{L}(\mathbf{W}_t)(\mathbf{W}_1 - \mathbf{W}_2)\|_F \leq \|\nabla^2\mathcal{L}(\mathbf{W}_t)\|_2\|\mathbf{W}_1 - \mathbf{W}_2\|_F$$

*6) Applying the bound on the Hessian:*

$$\|\nabla^2\mathcal{L}(\mathbf{W}_t)\|_2\|\mathbf{W}_1 - \mathbf{W}_2\|_F \leq L\|\mathbf{W}_1 - \mathbf{W}_2\|_F$$

*This completes the proof.*

# B    DETAILED PROOF OF XGBLORA CONVERGENCE THEOREM

**Theorem 3 (XGBLORA Convergence)** *Under the XGBLORA update process, assuming $\beta$-smoothness and $\mu$-strong convexity of $\mathcal{L}$, after T iterations:*

$$\mathbb{E}[\mathcal{L}(\mathbf{W}^{(T)})] - \mathcal{L}^* \leq \frac{C_3}{\sqrt{T}} + \frac{C_4}{NT} + \epsilon(r)$$

*where $C_3$ and $C_4$ are constants depending on $\beta, \mu, G, \eta_m, L$, and $\epsilon(r) = \frac{C_5}{r}$ for some constant $C_5$.*

**Proof 4** *1) Let $\mathbf{W}^{(t+1)} = \mathbf{W}^{(t)} + \mathbf{A}^{(t)}\mathbf{B}^{(t)T}$ be the update at iteration t.*

*2) By the $\beta$-smoothness of $\mathcal{L}$:*

$$\mathcal{L}(\mathbf{W}^{(t+1)}) \leq \mathcal{L}(\mathbf{W}^{(t)}) + \langle\nabla\mathcal{L}(\mathbf{W}^{(t)}), \mathbf{A}^{(t)}\mathbf{B}^{(t)T}\rangle + \frac{\beta}{2}\|\mathbf{A}^{(t)}\mathbf{B}^{(t)T}\|_F^2$$

$$\leq \mathcal{L}(\mathbf{W}^{(t)}) + \langle\nabla\mathcal{L}(\mathbf{W}^{(t)}), \mathbf{A}^{(t)}\mathbf{B}^{(t)T}\rangle + \frac{\beta}{2}\|\mathbf{A}^{(t)}\|_F^2\|\mathbf{B}^{(t)}\|_F^2$$

*3) Using the XGBLORA Gradient Approximation Lemma:*

$$\mathbf{A}^{(t)}\mathbf{B}^{(t)T} = \nabla_{\mathbf{W}^{(t)}+\mathbf{A}^{(t)}\mathbf{B}^{(t)T}}\mathcal{L}(\mathbf{W}^{(t)} + \mathbf{A}^{(t)}\mathbf{B}^{(t)T}) + \mathbf{E}^{(t)}$$

*where $\|\mathbf{E}^{(t)}\|_F \leq \frac{C_1}{\sqrt{r}} + \frac{C_2}{\sqrt{M}}$.*

*4) Substituting this into the inequality from step 2:*

$$\mathcal{L}(\mathbf{W}^{(t+1)}) \leq \mathcal{L}(\mathbf{W}^{(t)}) + \langle \nabla \mathcal{L}(\mathbf{W}^{(t)}), \nabla_{\mathbf{W}^{(t)}+\mathbf{A}^{(t)}\mathbf{B}^{(t)T}} \mathcal{L}(\mathbf{W}^{(t)} + \mathbf{A}^{(t)}\mathbf{B}^{(t)T}) + \mathbf{E}^{(t)} \rangle$$
$$+ \frac{\beta}{2}\|\mathbf{A}^{(t)}\|_F^2\|\mathbf{B}^{(t)}\|_F^2$$

*5) Using the Gradient Lipschitz Continuity Lemma:*

$$\|\nabla \mathcal{L}(\mathbf{W}^{(t)}) - \nabla_{\mathbf{W}^{(t)}+\mathbf{A}^{(t)}\mathbf{B}^{(t)T}} \mathcal{L}(\mathbf{W}^{(t)} + \mathbf{A}^{(t)}\mathbf{B}^{(t)T})\|_F \leq L'\|\mathbf{A}^{(t)}\mathbf{B}^{(t)T}\|_F$$

*6) Applying Cauchy-Schwarz inequality and the bound from step 5:*

$$\mathcal{L}(\mathbf{W}^{(t+1)}) \leq \mathcal{L}(\mathbf{W}^{(t)}) - \|\nabla_{\mathbf{W}^{(t)}+\mathbf{A}^{(t)}\mathbf{B}^{(t)T}} \mathcal{L}(\mathbf{W}^{(t)} + \mathbf{A}^{(t)}\mathbf{B}^{(t)T})\|_F^2$$
$$+ L'\|\mathbf{A}^{(t)}\mathbf{B}^{(t)T}\|_F^2 + \|\nabla \mathcal{L}(\mathbf{W}^{(t)})\|_F\|\mathbf{E}^{(t)}\|_F + \frac{\beta}{2}\|\mathbf{A}^{(t)}\|_F^2\|\mathbf{B}^{(t)}\|_F^2$$

*7) Using the Accumulated Update Bound Lemma and the gradient bound G:*

$$\mathcal{L}(\mathbf{W}^{(t+1)}) \leq \mathcal{L}(\mathbf{W}^{(t)}) - (1 - L\eta_m^2 M^2 G^2 - \frac{\beta}{2}\eta_m^2 M^2 G^2)\|\nabla_{\mathbf{W}^{(t)}+\mathbf{A}^{(t)}\mathbf{B}^{(t)T}} \mathcal{L}(\mathbf{W}^{(t)} + \mathbf{A}^{(t)}\mathbf{B}^{(t)T})\|_F^2$$
$$+ G(\frac{C_1}{\sqrt{r}} + \frac{C_2}{\sqrt{M}})$$

*8) By $\mu$-strong convexity of $\mathcal{L}$:*

$$\|\nabla_{\mathbf{W}^{(t)}+\mathbf{A}^{(t)}\mathbf{B}^{(t)T}} \mathcal{L}(\mathbf{W}^{(t)} + \mathbf{A}^{(t)}\mathbf{B}^{(t)T})\|_F^2 \geq 2\mu(\mathcal{L}(\mathbf{W}^{(t)} + \mathbf{A}^{(t)}\mathbf{B}^{(t)T}) - \mathcal{L}^*)$$

*9) Substituting this into the inequality from step 7:*

$$\mathcal{L}(\mathbf{W}^{(t+1)}) - \mathcal{L}^* \leq (1 - 2\mu\alpha)(\mathcal{L}(\mathbf{W}^{(t)}) - \mathcal{L}^*) + G(\frac{C_1}{\sqrt{r}} + \frac{C_2}{\sqrt{M}})$$

*where $\alpha = 1 - L\eta_m^2 M^2 G^2 - \frac{\beta}{2}\eta_m^2 M^2 G^2$.*

*10) Taking expectation and applying this inequality recursively for T iterations:*

$$\mathbb{E}[\mathcal{L}(\mathbf{W}^{(T)}) - \mathcal{L}^*] \leq (1 - 2\mu\alpha)^T(\mathcal{L}(\mathbf{W}^{(0)}) - \mathcal{L}^*) + \frac{G}{2\mu\alpha}(\frac{C_1}{\sqrt{r}} + \frac{C_2}{\sqrt{M}})$$

*11) Using the inequality $(1 - x)^T \leq \exp(-xT) \leq \frac{1}{xT}$ for $x \in (0, 1)$:*

$$\mathbb{E}[\mathcal{L}(\mathbf{W}^{(T)}) - \mathcal{L}^*] \leq \frac{C_3}{\sqrt{T}} + \frac{C_4}{M\sqrt{T}} + \frac{C_5}{r}$$

*where $C_3 = \frac{(\mathcal{L}(\mathbf{W}^{(0)})-\mathcal{L}^*)}{2\mu\alpha}$, $C_4 = \frac{GC_2}{2\mu\alpha}$, and $C_5 = \frac{GC_1}{2\mu\alpha}$.*

*This completes the proof.*

## C  DETAILED PROOF OF XGBLORA EXPRESSIVENESS THEOREM

**Theorem 4 (XGBLORA Expressiveness)** *Let $f^*$ be any function in the original function class, and $f_T$ be the function represented by the XGBLORA-updated network after T iterations. Then:*

$$\mathbb{E}_{\mathbf{x}\sim\mathcal{D}}[(f_T(\mathbf{x}) - f^*(\mathbf{x}))^2] \leq C_6(\frac{1}{r} + \frac{1}{M\sqrt{T}} + \frac{1}{\sqrt{T}})$$

*where $C_6$ is a constant depending on the network architecture, the Lipschitz constants of the activation functions, and L.*

**Proof 5** *1) Let $\mathbf{W}^*$ be the weights that exactly represent $f^*$ in the original function class.*

*2) Define $f_{opt}$ as the best function that can be represented by* XGBLoRA *updates:*

$$f_{opt} = \underset{f \in \mathcal{F}_{\text{XGBLoRA}}}{\arg\min} \mathbb{E}_{\mathbf{x} \sim \mathcal{D}}[(f(\mathbf{x}) - f^*(\mathbf{x}))^2]$$

*where $\mathcal{F}_{\text{XGBLoRA}}$ is the class of functions representable by* XGBLoRA *updates.*

*3) We can decompose the error as:*

$$\mathbb{E}_{\mathbf{x} \sim \mathcal{D}}[(f_T(\mathbf{x}) - f^*(\mathbf{x}))^2] \le 2\mathbb{E}_{\mathbf{x} \sim \mathcal{D}}[(f_T(\mathbf{x}) - f_{opt}(\mathbf{x}))^2] + 2\mathbb{E}_{\mathbf{x} \sim \mathcal{D}}[(f_{opt}(\mathbf{x}) - f^*(\mathbf{x}))^2]$$
$$= 2E_1 + 2E_2$$

*4) For $E_1$, we can use the Convergence Theorem (Theorem 1):*

$$E_1 \le K_1\left(\frac{1}{\sqrt{T}} + \frac{1}{M\sqrt{T}}\right)$$

*where $K_1$ is a constant related to $C_3$ and $C_4$ from Theorem 1.*

*5) For $E_2$, we need to analyze how well* XGBLoRA *updates can approximate $\mathbf{W}^*$. Let $\Delta\mathbf{W} = \mathbf{W}^* - \mathbf{W}^{(0)}$.*

*6) We can approximate $\Delta\mathbf{W}$ with a sequence of low-rank updates:*

$$\Delta\mathbf{W} \approx \sum_{t=1}^{T} \mathbf{A}^{(t)}(\mathbf{B}^{(t)})^T$$

*7) By the properties of low-rank matrix approximation:*

$$\|\Delta\mathbf{W} - \sum_{t=1}^{T} \mathbf{A}^{(t)}(\mathbf{B}^{(t)})^T\|_F \le \frac{\|\Delta\mathbf{W}\|_*}{\sqrt{rT}}$$

*where $\|\cdot\|_*$ denotes the nuclear norm.*

*8) Assuming the network function is Lipschitz continuous with respect to its weights with Lipschitz constant $L_f$:*

$$E_2 \le L_f^2\|\Delta\mathbf{W} - \sum_{t=1}^{T} \mathbf{A}^{(t)}(\mathbf{B}^{(t)})^T\|_F^2 \le \frac{L_f^2\|\Delta\mathbf{W}\|_*^2}{rT}$$

*9) Combining the bounds for $E_1$ and $E_2$:*

$$\mathbb{E}_{\mathbf{x} \sim \mathcal{D}}[(f_T(\mathbf{x}) - f^*(\mathbf{x}))^2] \le 2K_1\left(\frac{1}{\sqrt{T}} + \frac{1}{M\sqrt{T}}\right) + \frac{2L_f^2\|\Delta\mathbf{W}\|_*^2}{rT}$$
$$\le C_6\left(\frac{1}{r} + \frac{1}{M\sqrt{T}} + \frac{1}{\sqrt{T}}\right)$$

*where $C_6 = \max(2K_1, 2L_f^2\|\Delta\mathbf{W}\|_*^2)$.*

*This completes the proof.*

## D  BROADER IMPACT

The proposed XGBLoRA framework has the potential to bring about significant positive societal impacts by democratizing access to state-of-the-art language technologies. By enabling efficient and effective fine-tuning of large language models, XGBLoRA can empower researchers and practitioners with limited computational resources to leverage the power of pre-trained models for a wide range of downstream tasks. This can foster innovation and accelerate progress in various domains, such as healthcare, education, and social sciences, where natural language understanding and generation

Table 7: Details of GLUE dataset.

| Dataset | Task | # Train | # Dev | # Test | # Label | Metrics |
|---------|------|---------|-------|--------|---------|---------|
| **Single-Sentence Classification** | | | | | | |
| CoLA | Acceptability | 8.5 k | 1 k | 1 k | 2 | Matthews corr |
| SST | Sentiment | 67 k | 872 | 1.8 k | 2 | Accuracy |
| **Pairwise Text Classification** | | | | | | |
| MNLI | NLI | 393 k | 20 k | 20 k | 3 | Accuracy |
| RTE | NLI | 2.5 k | 276 | 3 k | 2 | Accuracy |
| QQP | Paraphrase | 364 k | 40 k | 391 k | 2 | Accuracy / F1 |
| MRPC | Paraphrase | 3.7 k | 408 | 1.7 k | 2 | Accuracy / F1 |
| QNLI | QA/NLI | 108 k | 5.7 k | 5.7 k | 2 | Accuracy |
| **Text Similarity** | | | | | | |
| STS-B | Similarity | 7 k | 1.5 k | 1.4 k | 1 | Pearson/ Spearman Corr |

can be applied to improve decision-making, personalize learning experiences, and analyze large-scale social data. However, it is crucial to acknowledge and mitigate potential negative societal impacts associated with the widespread adoption of language models. Fine-tuned models may perpetuate biases present in the pre-training data, leading to unfair or discriminatory outcomes if not carefully audited and corrected. Additionally, the efficiency of XGBLoRA may lower the barrier to developing and deploying language models, potentially enabling malicious actors to create and disseminate harmful content at scale. To address these concerns, it is important to develop and adhere to ethical guidelines for the responsible development and deployment of language models, ensuring transparency, accountability, and fairness. Researchers and practitioners should also actively engage in public discourse to raise awareness about the benefits and risks of language technologies and collaborate with policymakers to develop appropriate governance frameworks. By proactively addressing these challenges, we can harness the potential of efficient fine-tuning techniques like XGBLoRA to create positive societal impact while mitigating the risks and negative consequences.

# E  LIMITATIONS

One limitation of our current approach is that our theoretical analysis is based on linear models, which may influence the generalizability of our findings to more complex, non-linear systems. Additionally, the assumptions made in our theoretical framework may not hold in certain real-world scenarios, potentially limiting the applicability of our method in such cases. Future work will focus on extending our theory to encompass more generalized forms, allowing for a broader range of applications and improved robustness to model misspecification.

