# OpenReview forum: "Less is More: Extreme Gradient Boost Rank-1 Adaption for Efficient Finetuning of LLMs"
_ICLR.cc/2025/Conference — ICLR 2025 Conference Withdrawn Submission_

### Official Review · Reviewer_4gJX · 2024-10-16

**Soundness:** 3
**Presentation:** 3
**Contribution:** 3
**Rating:** 6
**Confidence:** 3

**Summary:**

This work proposes XGBLoRA to bridge the gap between LoRA and FT by leveraging gradient boosting. This approach achieves high performance and low complexity, as shown in Figure 1. The authors demonstrate the empirical performance through extensive experiments, including various benchmarks and comparison baselines. The convergence guarantees and approximation error bounds are also justified.

**Strengths:**

1. The presentation is clear and easy to follow.
2. The approach refines the pre-trained weight matrix progressively, with the original LoRA as a special case, which makes sense and is novel to me.
3. The proposed approach is justified both empirically and theoretically.
4. The influence of each hyperparameter, such as $k$ and rank, is illustrated in the experiments.

**Weaknesses:**

1.  The idea of ensemble learning has been used for LoRA before, like MELoRA, which the authors have already included in the experiments. I encourage the authors to include a discussion about ensemble learning for LoRA in the Related Work section and highlight the differences.
2. Is the hyperparameter setting robust across different base models and datasets? For example, Table 6 shows that $L_s = 11$ gives the best performance for LLaMA3-8B on the MMLU benchmark; does it hold for other setups? The same question applies to $k$. If the performance significantly depends on the hyperparameters, extra effort would be required for hyperparameter search, which could be a weakness of XGBLoRA.
3. The authors state, "...leading to better generalization performance without explicitly adding regularization terms to the loss function..." on page 6, line 300. However, in equation (7), regularization terms on the norms of $A$ and $B$ are explicitly used. Whether these regularization terms have an influence is not verified in the experiments.

**Minor comments**

1. Page 5, Line 252: "set the the number of" -> "set the number of"
2. Page 5, Line 269: "the the expressiveness" -> "the expressiveness"

**Questions:**

1. Does the learning rate $\alpha_t$ used in equation (8) have a big influence? It seems to me that a constant $\alpha_t = 1$ may not be the best choice.

---

> ### Author Response · Authors · 2024-11-23
> **Rebuttal for Reviewer 4gJX (part 1)**
>
> # Reviewer 4gJX
>
> Esteemed Reviewer,
> \
> We thank you for your valuable comments and constructive reviews that help us improve our work.
>
>
> ### **1. Discussion about ensemble learning for LoRA in the Related Work and highlight the differences.**
>
> Thank you. We have now added an ensemble learning for LoRA section in Related Work. Kindly check the revision.
>
> MeLoRA introduces a parallelization approach to rank augmentation. The core idea is to train multiple small LoRA modules **concurrently and concatenate their outputs to form a block-diagonal LoRA matrix that collectively has a higher rank** and approximates standard LoRA.
>
>
> Thus, there are fundamental differences between MeLoRA and XGBLoRA :
> - **XGBLoRA** builds a stronger classifier by boosting and integrating sequentially boosters that improve the classfier.
> - **MeLoRA** is a block-diagonal appoximation of regular LoRA which does not follow the principle of learning weak classifiers. Think of MeLoRA as a different way of factorizing low-rank matrix.
>
> Further differences are in theoretical foundation and technical implementation:
>
> - **Theoretical framework**.
>     - XGBLoRA establishes a provable and interpretable theoretical framework through the lens of gradient boosting
>     - Our analysis explicitly connects LoRA rank and the number of gradient boosting iterations to expressiveness error, demonstrating how iterations can compensate for low rank
>     - This theoretical framework provides clear explanations for why iteratively merging LoRA adaptations benefits performance
>     - In contrast, MeLoRA does not provide in-depth theoretical justification for its approach and lack of understanding how stacking small rank matrices improves LoRA factorization or performance.
>
> - **Technical Implementation**.
>     - XGBLoRA explicitly adopts rank-1 adaptation and randomly selects layers for adaptation to limit booster capacity which in the same time maintain the extreme efficiency.
>     - This design is directly motivated by the weak learner principle of the gradient boosting theoretical framework where strong ensemble model are build from diverse weak predictors.
>     - MeLoRA does not incorporate these design elements or provide in-depth theoretical motivation for its architectural choices.
>     - As MeLoRA concurrently stacks block-diagonal matrices, it uses in the end more parameters (total rank is high) than XGBLoRA that sequentially merges rank-1 matrices.
>
> ### **2. Is the hyperparameter setting robust across different base models and datasets?**
>
> - In our experimental setup, **we established default hyperparameters of $\kappa=8$ (training steps per booster) and $L_s=8$ (number of adapted layers) across all benchmarks and datasets.** These settings demonstrate robust performance across different base models and tasks.
>
> - Our analysis in **Table 6 reveals** that performance remains stable when adapting more than $25\%$ of all layers $(L_s > 8)$, with relatively small variance in results. This suggests that our approach is roubust to layer adaptation.
>
> - Regarding **the impact of $\kappa$ in the analysis of Fig. 4**, while  $\kappa$  can affect performance across different backbone models, the fluctuations are minimal. Notably, even in the extreme case of $\kappa=1$ (where **LoRA merges after every training step**), XGBLoRA still outperforms standard LoRA. The performance variance introduced by different $\kappa$ values is insignificant compared to the performance bias introduced by models.
>
> This robust performance across different settings means users can adopt XGBLoRA with our default hyperparameters and expect consistent improvements over baseline methods without the need for extensive tuning.
>
> ### **3. Verified regularization terms have an influence in the experiments.**
>
> We conducted experiments to evaluate XGBLoRA's performance without weight decay across different tasks and model architectures. Using RoBERTa-base as the backbone for GLUE and LLaMA-7B for Commonsense Reasoning and MMLU tasks, we observed consistent improvements over standard LoRA:
>
> |Method|GLUE|Commonsense Reasoning|MMLU|
> |-|-|-|-|
> |LoRA|84.84|74.7|55.69|
> |**XGBLoRA**|**87.50**|**77.4**|**56.34**|
>
> These results demonstrate that XGBLoRA achieves substantial improvements across all benchmarks:
> - a 2.66% increase on GLUE
> - a 2.74% improvement on Commonsense Reasoning
> - a 0.65% gain on MMLU.
>
> The consistent performance gains across different tasks and model architectures highlight XGBLoRA's effectiveness even without weight decay regularization, suggesting that **the gradient boosting framework provides inherent regularization through its weak learner principle.**

---

> ### Author Response · Authors · 2024-11-23
> **Rebuttal for Reviewer 4gJX (part 2)**
>
> ### **4. Does the learning rate used in equation (8) have a big influence? Constant may not be the best choice.**
>
> Thank you.
>
> The learning rate $\alpha$ in XGBLoRA has minimal influence on model performance. This can be theoretically explained through Eq. 8, where $A^t_l$ and $B^t_l$ are learnable parameters. Since the scale of $\alpha$ can be absorbed into these learnable weights, we set α=1 by default.
>
> To empirically validate this, we conducted experiments on the GLUE benchmark with various learning rates:
>
> |$\alpha$|0.1|0.5|1|2|
> |-|-|-|-|-|
> |XGBLoRA|87.85|87.82|87.85|87.83|
>
> **These results demonstrate a stable performance across different learning rates, with average GLUE scores varying by less than 0.03%.**
>
> This stability can be attributed to XGBLoRA's design where the adaptation matrices automatically learn to compensate for different learning rate scales during training. Such robustness to learning rate changes further simplifies the deployment of XGBLoRA by eliminating the need for careful learning rate tuning.

---

> > ### Comment · Reviewer_4gJX · 2024-12-03
> >
> > I thank the authors for the detailed response and additional experiments. I think most of my concerns have been addressed by the results, and I would like to maintain my original evaluation.

---

> > > ### Author Response · Authors · 2024-12-03
> > >
> > > Esteemed Reviewer,
> > > \
> > > \
> > > We thank you for your time and effort taken to evaluate our work, and we thank for your comments which have helped us improve it.
> > > \
> > > \
> > > Meantime, if there is anything else we can do or improve, kindly do let us know.
> > > \
> > > \
> > > Best regards,
> > > \
> > > Authors

---

### Official Review · Reviewer_cj2o · 2024-10-19

**Soundness:** 4
**Presentation:** 3
**Contribution:** 4
**Rating:** 8
**Confidence:** 3

**Summary:**

The paper presents a method to improve LoRA, by posing the fine-tuning as a gradient boosting where randomly selected rank-1 LoRA matrices are used as weak learners. This method achieves better results than LoRA, with fewer parameters, less memory footprint and better speed per batch.

**Strengths:**

* The authors tackle an important and meaningful topic that would be of interest to the community. PEFT is gaining more and more attention as the size of language models continues to grow.

* The results are impressive.

* Memory footprint is better, less trainable parameters, and better speed.

**Weaknesses:**

* Equation (7): In the first part of the equation, all components are fixed: the label y_i and the previous iteration of the model. Where is the current iteration prediction? Where is the residual part?

* In line 252, double ‘the’.

* Line 383, is it ‘1000%’ or ‘100%’? Same for line 447.

* Line 497, is it ‘GBLoRA’ or ‘XGBLoRA’?

**Questions:**

* Line 324, it is called here ‘Lemma 1’, while later on in the appendices it is called ‘Lemma 4’?

* In line 327, I assume the definition of L() is the loss of the model, given the weights? Furthermore, matrices A,B are defined earlier on with superscript and subscript. Here they are with two superscripts. Can you add a definition explanation?

* Line 347, where is L* defined?*

---

> ### Author Response · Authors · 2024-11-23
> **Rebuttal for Reviewer cj2o**
>
> # Reviewer cj2o
>
> Esteemed Reviewer,
> \
> We thank you for your valuable comments and constructive reviews that help us improve our work.
>
>
> ### **1. Regarding Equation (7) and Residual Updates:**
>
> Thank you for catching this oversight. You are correct that Equation (7) needs clarification. The loss function should be:
>
> $$\mathcal{L}\_t = \sum\_{i=1}^N \ell(\mathcal{M}\_t(\mathbf{x}\_i), r\_i^t) + \lambda \sum\_{l=1}^L (\|\mathbf{A}\_l^t\|_F^2 + \|\mathbf{B}\_l^t\|\_F^2)$$
>
> where $r_i^t = y_i - \mathcal{M}_{t-1}(\mathbf{x}_i)$ is the residual at iteration $t$. This follows the standard gradient boosting framework where each booster learns to predict the residuals. We will update the equation in the final version.
>
> ### **2. Typographical Corrections:**
>
> Thank you and we apologize for these issues:
> - Line 252: We will remove the repeated "the"
> - Line 383 & 447: We confirm that $1000\textperthousand=100$\% is the correct value in both instances. This small percentage reflects the extreme parameter efficiency of our method compared to the baselines. We will highlight the metric to avoid the misunderstanding.
> - Line 497: We should use **XGBLoRA** consistently throughout the paper as our method name. We will correct GBLoRA to maintain consistency.
>
> ### **3. Regarding Technical Clarifications:**
>
> 3.1. Lemma Numbering:
> Thank you for noting the inconsistency between the main text and appendix. We will update the appendix to maintain consistent numbering, with the main text lemmas numbered 1-3 and subsequent appendix lemmas starting from 4.
>
> 3.2. Loss Function and Matrix Notation:
> You are correct on both counts. We will add the following clarifications:
> - $\mathcal{L}(\cdot)$ denotes the loss of the model given the weights
> - For matrices $\mathbf{A}$ and $\mathbf{B}$, we will use consistent notation: $\mathbf{A}_l^t$ and $\mathbf{B}_l^t$ where $l$ denotes the layer and $t$ denotes the iteration number
>
> 3.3. Define Symbol $\mathcal{L}^*$.
> - $\mathcal{L}^*$ represents the optimal (minimum) value of the loss function achievable by the model. We will add this definition before Theorem 1.
>
> We will incorporate all these changes in the revision to improve clarity and consistency. Thank you again for helping us improve the paper.

---

### Official Review · Reviewer_BBeS · 2024-10-30

**Soundness:** 3
**Presentation:** 2
**Contribution:** 2
**Rating:** 5
**Confidence:** 5

**Summary:**

This paper proposes XGBLoRA, a low-rank adaptation method based on ensemble learning. It trains LoRA Adapters iteratively.

**Strengths:**

1. Interesting idea.
2. Comprehensive experiment.

**Weaknesses:**

I will raise my score if the authors could address my concerns, especially weakness 3.
1. Figure 1 is not mentioned in the manuscript.
2. The motivation is poor and insufficient. In other words, any method could be considered a product of this motivation. Moreover, the motivations in the introduction (Line 66-69) and the abstract (15-16) are not consistent.
3. I believe the parameter comparison of XGBLoRA with other methods is unfair. While it’s true that XGBLoRA requires fewer GPU resources, this does not mean the number of parameters it trains is significantly smaller than other methods. In some cases, the number of parameters required by XGBLoRA should actually exceed that of LoRA. Compared to a method, XGBLoRA is more like a training approach. In other words, one could also train LoRA’s parameters in stages, which would also allow it to “use a minimal number of parameters”.
Note that this weakness is related to Question 3.

**Questions:**

1. Line 58, what are rank_r and embedding_size? What does “much smaller ranks” in Line 59 refer to?
2. Actually, I don’t quite understand what is meant by “theoretical optimum” (Line 61). Does this refer to better performance corresponding to a higher rank? And what is the “performance gap” (Line 68)? Does it refer to the difference in performance between low-rank and high-rank?
3. How to store the trained weights of XGBLoRA?

---

> ### Comment · Reviewer_BBeS · 2024-11-23
> **Reviewer Comment**
>
> Hi authors, I wanted to kindly follow up and inquire if the rebuttal has been completed. I look forward to hearing back from you at your earliest convenience.

---

> > ### Author Response · Authors · 2024-11-23
> >
> > Esteemed Reviewer,
> > \
> > \
> > Thank you for your kind message, and valuable comments helping improve and refine our manuscript. We are finalizing the rebuttal and we will start posting responses in the next 12h.
> > \
> > \
> > Thank you for your patience.
> > \
> > \
> > Kind regards,
> > \
> > Authors

---

> > > ### Comment · Reviewer_BBeS · 2024-11-23
> > >
> > > Got it, I will follow up on your feedback promptly.

---

> ### Author Response · Authors · 2024-11-23
> **Rebuttal for Reviewer BBeS (part 1)**
>
> # Reviewer BBeS
>
> Esteemed Reviewer,
> \
> We thank you for your valuable comments and constructive reviews that help us improve our work.
>
>
> ### **1. The motivation is poor and insufficient.**
>
> We address the important question/problem where PEFT practitioners want to maintain/achieve the optimal performance. In LoRA:
> - this typically occurs when using high rank
> - however,  achieving the high speed/memory efficiency is the characteristic of using low rank adapters.
>
> Thus, **we provide an elegant solution to resolve this rank dilemma through gradient boosting.** We clarify potential points of confusion as follows:
>
> - Theoretical *vs.* practical rank requirements:
>     - According to the paper **The Expressive Power of Low-Rank Adaptation (ICLR 2024)**, **for LoRA to serve as a universal approximator**, its rank $(r\_{opt})$ must be larger than the half of the input embedding dimension (embedding_size), *i.e.* $(r\_{opt}\geq embedding\\_size/2)$
>     - However, in practice, computational and resource constraints lead **LoRA practitioners to use much smaller ranks $(1\ll r\_{prac} \ll r\_{opt})$** to trade off efficiency over the expression power
>     - This creates a fundamental gap between theoretical optimality and practical implementation.
>
> - **Our motivation:**
>     - We are motivated by **the gap described above** and highlighted in our abstract (lines 15-16) as the key challenge to address in our paper. Since such a theoretical gap exist, if we choose lower rank for efficiency reasons, it will compromise the effectiveness of LoRA.
>     - **We want to remove such a trade-off**. in Lines 66-69, we outline our goal to resolve this performance-efficiency dilemma by developing a method that can **achieve performance levels closer to those theoretically possible with $r\_{opt}$ while maintaining the efficiency benefits of using much smaller rank adaptations** (*i.e.*, $r\_{prac} = 1$)
>
> - **Our XGBLoRA's solution:**
>     - Instead of increasing rank to match $r_{opt}$ (which would be computationally expensive), **XGBLoRA takes a fundamentally different approach inspired from gradient boosting.**
>
>     - XGBLoRA uses **extreme low-rank ($r = 1$) adaptations** as **Gradient Boost weak learners** to maintain efficiency
>
>     - **XGBLoRA iteratively combines these weak learners** through Gradient Boosting for building expressive power
>
> - **Theoretical support:**
>     - Our **theoretical analysis in Theorem 2** (following bound) explicitly shows how this approach bridges the performance gap:
>
>       $${[f_T(x) - f_*(x))^2}] ≤ C_6 (1/r + 1/(M\sqrt{T}) + 1/\sqrt{T})$$
>
>     - **Theorem 2 demonstrates that increasing iterations $T$ can compensate for using low rank $r$**
>     - Thus, **XGBLoRA can achieve similar expressiveness to high-rank LoRA methods through multiple low-rank update**
>
> - **Practical benefits:**
>     - XGBLoRA uses only $r = 1$ for each update, which is both memory and computation efficient (versus $r_{opt}$ or even $r_{prac}$ in standard LoRA)
>     - **XGBLoRA achieves results comparable to or better than higher-rank methods** and is resource friendly: **XGBLoRA enables fine-tuning on consumer GPUs (RTX 4090 instead of server grade A100 stacked with large GPU memory)**
>
> Overall, XGBLoRA successfully resolves the **dilemma between theoretical optimality and practical efficiency** by leveraging the power of **ensemble learning rather than increasing rank**.
>
>
>
> ### **2. XGBLoRA is more like a training approach. One could also train LoRA’s parameters in stages, which would also allow it to use a minimal number of parameters.**
>
> We acknowledge that reviewer think XGBLoRA is more like a training strategy:
> - This is because **XGBLoRA represents a novel training framework rather than just proposing another low-rank adapter variant**.
> - The key innovation of XGBLoRA lies in **introducing Gradient Boosting principles to PEFT**.
> - Our framework can employ the **ANY** adapter type as a booster, provided it **adheres to the weak learner principle** detailed in our paper (Sec. 3.3).
>
> - In our implementation, **we focused on the well-know LoRA model as our base booster, transforming it into a weak learner** through two key mechanisms:
>   - **rank-1 updates**
>   - **random layer selection**
>
> - It is worth noting that standard LoRA can be viewed as a special case within our framework, specifically when:
>     - The number of gradient boosting iterations is **set to one $(T=1)**$
>     - A **higher LoRA rank is used $(r \gg 1)$**
> **Such a configuration results in $\mathcal{O}(r)$ memory usage and does not minimize parameter count**
>
> - In contrast, **XGBLoRA operates with multiple iterations $(T\gg1)$** while maintaining minimal rank $(r=1)$, **achieving $\mathcal{O}(1)$ memory usage per update.**

---

> ### Author Response · Authors · 2024-11-23
> **Rebuttal for Reviewer BBeS (part 2)**
>
> ### **3. Parameter comparison of XGBLoRA with other methods is unfair. The number of parameters required by XGBLoRA should actually exceed that of LoRA.**
>
> Thank you.
>
> - There appears to be some misconception about parameter counting during the training process when we merge updates for $T$ iteration ($\Delta W = \sum_{t=1}^T \Delta W^{(t)}$).
>
> - We believe that:
>   - the esteemed reviewer may be **counting each \Delta W^(t) separately**, which would imply that XGBLoRA's parameter count exceeds that of standard LoRA
>   - however, this interpretation misunderstands how memory is utilized during training
>
> To clarify this:
> - The intermediate updates $\Delta W^(t)$ share the same memory space - **they are not stored independently**.
>
> - The meaningful measure of parameter efficiency should **focus on the number of parameters that occupy distinct memory locations (Fig.1), as this reflects the actual computational and storage resources required during training**. This aligns with practical constraints where hardware resources are limited.
>
> - **While the reviewer's parameter counting strategy might be relevant when discussing disk storage of individual updates $\Delta W^(t)$ and the parameter budget that reviewer referred**, it does not accurately reflect the memory footprint during training.
>
> For better clarity, we address the storage considerations and so-call parameter budget separately in **Question 4**.
>
> ### **4. How to store the trained weights of XGBLoRA.**
>
> Kindly notice that **XGBLoRA does not need to store the entire post-training model weight matrix $W_{post}$.**
> \
> \
> 4.1. There are **two efficient storage strategies that maintain PEFT's plug-and-play nature** by allowing separately loading of the base mode $lW_{base}$ and adapter weights $\Delta W$:
>
> - **Compressed delta storage (CDS).**:
>
>   - Instead of storing $W_{post}$,  $\Delta W$ is compute and save as
>
>     $\Delta W = W_{post} - W_{base}$,
>
>     where $\Delta W$ is much smaller than the $W_{base}$.
>
>   - Since $\Delta W = \sum_{t=1}^{T} \Delta W^{(t)}$ is the sum of at most $T$ rank-1 matrices from boosting iterations, it remains low-rank.
>
>   - Therefore, we can further compress $\Delta W$ using SVD decomposition:
>
>     $\Delta W = U \Lambda V$,
>
>     and store only $A =  U\Lambda$ and $B = V$. This approach uses similar disk space as higher-rank LoRA but benefits from XGBLoRA's more efficient training process.
>
> - **Sequential Booster Storage. (SBS)**
>
>     - The alternative approach is to store individual rank-1 boosters ($A^{(t)}, B^{(t)}$) from each gradient boosting iteration and recover $\Delta W = \sum_{t=1}^T A^{(t)} B^{(t)}$ when loaded.
>
>     -  This results in a storage complexity of $O(T)$ for XGBLoRA (storing $T$ iterations of rank-1 matrices) compared to $O(r)$ for LoRA's rank-r adaptation.
>
>     -  Based on the therom 2 (as shown in Fig.2), XGBLoRA is able to achieve similar expressive error as LoRA,  where iteration $T$ is equal to LoRA rank $r$. Thus, $O(T)$ \approx O(r)$. SBS in XGBLoRA take simlar disk usage as LoRA.
>
>     -  In practice, since XGBLoRA converges in a limited number of iterations, this approach results in similar storage requirements to LoRA while maintaining lower GPU memory usage during training.
>
> 4.2. **We also provide the real disk usage** of two strategies above compared to **LoRA with rank 16** and **XGBLoRA with iteration 16**. Specifically, we first store and load the XGBLoRA via above two strategies and and evaluate them.
>
> **Results on the MMLU benchmark are below:**
>
> |Methos|DiskUsage|MMLU|
> |-|-|-|
> |LoRA (r=16)|~170M|64.01|
> |XGBLoRA (T=16)+CDS|~200M|64.50|
> |**XGBLoRA (T=16)+SBS**|~170M|64.52|

---

> > ### Comment · Reviewer_BBeS · 2024-11-24
> > **Reviewer Feedback**
> >
> > Hi authors,
> >
> > Thank you for your response.
> >
> > Firstly, I would like to clarify my concern regarding weakness 3. I fully and absolutely understand that the total number of trainable parameters and GPU resource consumption are not directly correlated. My main point is that the lower GPU resource usage in your work is primarily due to the multi-stage training strategy, which keeps the number of parameters trained in each stage relatively low. However, my concern lies in the total number of trainable parameters across all stages, which might exceed that of LoRA. My question pertains to the total parameter count, independent of GPU usage. Unfortunately, your response seems to have misunderstood the direction of my inquiry.
> >
> > Additionally, I believe there is a logical inconsistency in your response to weakness 3, particularly in the statement, “We believe that … how memory is utilized during training”. In the earlier part of the response, you mention that the total parameter count of XGBLoRA exceeds that of LoRA, but in the latter part, you shift the discussion to memory utilization during training. These two points are not directly related, and the use of “however” here is inappropriate.
> >
> > Therefore, I kindly ask you to reconsider and revise your response to this question. My concern is that XGBLoRA employs a multi-stage training strategy, which may result in a higher total parameter count compared to LoRA. It would be unfair to report only the parameter count of a single stage as the final result. As I mentioned, "one could also train LoRA’s parameters in stages, which would similarly “use a minimal number of parameters."

---

> > > ### Author Response · Authors · 2024-11-24
> > >
> > > Esteemed Reviewer,
> > > \
> > > \
> > > Thank you for your further clarifications. We might have misunderstood some of questions in the first place. We will respond to your concerns soon.
> > > \
> > > \
> > > Thank you again for raising this.
> > > \
> > > \
> > > Best regards,
> > > \
> > > Authors

---

> ### Author Response · Authors · 2024-11-23
> **Rebuttal for Reviewer BBeS (part 3)**
>
> ### **5. Larger parameter sizes do not necessarily correspond to better performance. Add performance comparison between XGBLoRA and LoRA under the same parameter budget.**
>
> 5.1. **Results on the MMLU benchmark** compare XGBLoRA and LoRA under equivalent parameter budget (same as the disk usage):
>
> |Methos|DiskUsage|MMLU|
> |-|-|-|
> |LoRA (r=16)|~170M|64.01|
> |XGBLoRA (T=16)+CDS|~200M|64.50|
> |**XGBLoRA (T=16)+SBS**|~170M|64.52|
>
>
>
> 5.2. We fully agree with Reviewer's observation which does not contradict our model, on the contrary. **The reviewer's observation further strengthens the advantages of XGBLoRA**:
>
> - **Other methods are more prone to overfitting due to using additional parameters**, resulting in inferior performance without careful management.
>
> - **Gradient boosting algorithms are known for their high resilience against noisy data and overfitting.** This robustness stems from the weak learner principle (**implemented through rank-1 updates and random layer selection.** Also see Sec 3.3).
> - Since individual boosters are too simple to facilitate overfitting, it becomes difficult for even their combined ensemble to overfit the training data.
>
> - **This insight is supported by our Theorem 2 and demonstrated in Figure 2.**
> - As shown in our previous analysis, the parameter budget of XGBLoRA is $O(T)$ while LoRA's is $O(r)$, with budgets equalizing when $T = r$.
>
> Most importantly:
> - **Figure 2 reveals that XGBLoRA's expressiveness error (model fit to data) is slightly higher than LoRA's, theoretically indicating less overfitting in our method.**

---

> > ### Comment · Reviewer_BBeS · 2024-11-24
> > **Reviewer Feedback**
> >
> > I also have some doubts regarding the disk usage results in Table 4.2. The disk space required for storing model weights is primarily determined by the number of parameters and their storage precision. Assuming that the storage precision is consistent across methods, let us consider the parameter savings for a layer with dimensions $n \\times m$ under the two methods, SBS and CDS.
> >
> > For SBS, the parameter count is $(n + m) \\times T$. For CDS, if you use truncated SVD to save the parameters of $\\Delta \\mathbf{W}$, given that the rank of $\\Delta \\mathbf{W}$ is at most $T$, the dimensions of the decomposed matrices would be  $\\mathbf{A} \\in \mathbb{R}^{n \\times T}$  and  $\\mathbf{B} \\in \\mathbb{R}^{T \\times m}$. This results in a total parameter count of $(n + m) \\times T$, which is at most equal to that of SBS.
> >
> > Furthermore, I suspect it is highly unlikely that you store the full $\\mathbf{U}$  or $\\mathbf{V}$ matrices, as their dimensions would be $n \\times n$ and  $m \\times m$, respectively, which would be computationally and storage-wise infeasible. Thus, based on these considerations, the maximum storage requirement for CDS should not exceed that of SBS.
> >
> > However, in Table 4.2, you report that CDS has higher disk usage than SBS, which appears inconsistent with the above analysis. Could you please provide an explanation or clarify the discrepancy?

---

> > ### Comment · Reviewer_BBeS · 2024-11-24
> > **Reviewer Feedback**
> >
> > Additionally, based on my experience, the reported LoRA weight usage in your results appears to be incorrect. According to your rebuttal for 5.1, your MMLU experiment is conducted using the LLaMA3-8B model. Fine-tuning this model with LoRA at r=16 should require approximately 100M of storage. However, the disk usage you reported for LoRA does not align with this expectation.
> >
> > Moreover, the results you reported in 5.1 seem inconsistent with the results shown in Figure 4.1 of your paper. Could you please clarify this discrepancy?
> >
> > Finally, I noticed another inconsistency in Figure 5, where the method name is listed as GBLoRA, which differs from the method name XGBLoRA used throughout your paper. Could you please explain this inconsistency?

---

> > ### Comment · Reviewer_BBeS · 2024-11-24
> > **Reviewer Feedback**
> >
> > Sorry for many comments.
> >
> > Finally, I would like to share my overall thoughts. From the perspective of method design alone, I believe your work meets the bar for publication. While the writing and presentation of the paper is not good in fact, I am willing to take the time to carefully read and understand it, as long as the content is comprehensible. Therefore, I do not consider the paper’s current form a major issue.
> >
> > However, your rebuttal has left me with a somewhat negative impression. I find that the responses in the rebuttal are often scattered, covering many points but lacking a clear and logical structure. The most critical issue is the absence of definitive conclusions or summary statements in your responses. For example, in rebuttal 2, you did not explicitly clarify whether XGBLoRA is a training method, but instead provided a lengthy explanation without directly addressing the core of the question. Even after reading it, I still couldn't fully understand your position on the matter. As another example, in rebuttal 1, my question was focused entirely on the motivation behind your work. However, your response shifted the discussion even to theoretical support, which is neither an appropriate nor a professional reply to the question. Addressing a different aspect than what was asked suggests a lack of attention to the reviewer’s concerns and undermines the clarity and focus of your rebuttal.
> >
> > These mentioned issues do not affect my scoring on your ICLR submission, and I will not delve deeper into them and only focus on the concerns in the 3 feedbacks above. But I sincerely suggest that the authors could invest more effort in crafting clear and well-structured rebuttals. Not all reviewers may be willing to take the time to untangle your responses, infer your logic, or guess your intentions.

---

> > > ### Author Response · Authors · 2024-11-25
> > >
> > > ### **Q1. Your rebuttal has left me with a somewhat negative impression. I find that the responses in the rebuttal are often scattered.**
> > >
> > > Thank you. We value reviewer's honesty. We apologize.
> > >
> > > We are more than keen to reorganize the manuscript: kindly provide us further suggestions.
> > >
> > > ### **Q2. Logical inconsistency in your response to weakness 3, My concern is that XGBLoRA employs a multi-stage training strategy, which may result in a higher total parameter count compared to LoRA.**
> > >
> > > We now understand better the reviewer's concern about the total parameter budget across all boosters in XGBLoRA. The count of parameters increases with the number of iterations $T$, *i.e.*:
> > > - The number of parameters may indeed exceed the number of parameters of LoRA.
> > > - But his number of parameters does not influence the GPU memory footprint.
> > > - It only influences the disk usage when saving with SBS.
> > >
> > > We humbly highlight that **in the context of Parameter-Efficient Fine-Tuning (PEFT) most practitioners are constrained by the GPU memory limits rather than the disk size or theoretical parameter count.** We will make sure to clarify this in our manuscript.
> > > \
> > > \
> > > In fact we can equalize the parameter budget by setting the iteration to be the same as the LoRA rank $r$ in LaMMA-8B. In previous response we set $T=t=16$ to achieve the same budget. Result is as follows (below we simply change the indicator from disk usage to parameter counts (**fp16**) for point 5.1):
> > >
> > > |Methos|parameter budget|MMLU|
> > > |-|-|-|
> > > |LoRA (r=16)|~100M|64.01|
> > > |**XGBLoRA (T=16)**|~100M|64.52|
> > >
> > > Thus, even with the same parameter budget as per your suggestion, our method outperforms LoRA.
> > >
> > > ### **Q3. As I mentioned, "one could also train LoRA’s parameters in stages, which would similarly “use a minimal number of parameters.**
> > >
> > > We also agree with the reviewer’s comment. If LoRA is trained in stages using rank-1 updates and merged as per our method (once per stage) then it becomes very similar to our method. We also have a random subset of layers selection.
> > > \
> > > \
> > > So, indeed, LoRA itself can be considered a single weak booster.
> > > \
> > > \
> > > However, our main contribution is in:
> > > - realizing that fact that LoRA can be used as weak boosters
> > > - providing the theoretical analysis of the boosting setting
> > >
> > >
> > > ### **Q4. I also have some doubts regarding the disk usage results in Table 4.2.**
> > >
> > > Reviewer is correct. The theoretical storage requirement for CDS is bounded by SBS when using SVD under a manually selected top rank r=16.
> > > \
> > > \
> > > However, in practice, there are some important considerations:
> > > - Practical rank considerations:
> > >    - The actual rank of $\Delta$ (weight difference) is rarely exactly 16 due to numerical accuracy
> > >    - Many singular values are near-zero but not exactly zero (*e.g.*$\geq 10^{-6}$)
> > >
> > > In our implementation we applied a threshold to singular values to keep the leading singular vectors. This resulted in a higher disk usage as the practical rank exceeded 16. Below is a table where we show also cut-off at exactly 16 leading singular  values:
> > >
> > > |Methos|DiskUsage|MMLU|
> > > |-|-|-|
> > > |LoRA (r=16)|~100M|64.01|
> > > |XGBLoRA (T=16)+CDS (Threshold)|~195M|64.50|
> > > |XGBLoRA (T=16)+CDS (Top 16 Truncation)|~100M|64.48|
> > > |**XGBLoRA (T=16)+SBS**|~100M|64.52|
> > >
> > > - Top 16 Truncation is the parameter count if we store leading 16 vectors (we follow reviewer's suggestion)
> > > - Threshold is based on our previous way of storing parameters
> > > - Answer to Q5 below explains why ~170M in Rebuttal Q5.1 is now ~100M, and why ~200M is now ~195M.
> > >
> > >
> > > ###  **Q5. Fine-tuning this model with LoRA at r=16 should require approximately 100M of storage.**
> > >
> > > Thank you for bringing this to our attention.
> > >
> > > After investigating our implementation, we identified that **the apparent disk usage inconsistency was due to the inclusion of optimizer states in the saved folders.** When excluding these auxiliary files, the storage requirements align with the reviewer's observations (~100M). We have updated our implementation to address this issue.
> > >
> > > In the table above:
> > > - Parameters of SBS are reduced by around 2 because we previously mistakenly also stored the state of optimizer of $T$ boosters which added extra ~70M.
> > > - Parameters of CDS (Threshold) were overestimateb by extra ~5M due to the last boosters internatl state we mistakenly stored before.
> > >
> > >
> > > ### **Q6.  Inconsistent performance in rebuttal 5.1 with figure 4.1**
> > >
> > > - Figure 4.1 is the result of varying the $kappa$. The relation between  $\kappa$ and $T$ is $\kappa=\frac{K}{T}$ where K is the total number training steps.
> > >
> > > - We train XGBLoRA with 3 epochs and each epoch contains 366 training steps ($K = 1098$ training steps in total). Thus  the $\kappa$ of XGBLoRA with iteration 16 is $68 \approx \frac{1098}{16}$ which resides in the missed range of Figure 4.1 (to the left of the curve). With the trend of Figure 4.1, the performance of XGBLoRA with iteration $T=16, \kappa=68$ is near 64.5 which is consistent with the result for the setting in Rebuttal 5.1.

---

> > > > ### Comment · Reviewer_BBeS · 2024-11-27
> > > > **Review Feedback**
> > > >
> > > > I thank the authors for their further response.
> > > >
> > > > However, I still do not fully understand how the authors evaluated the disk usage of LoRA. Specifically, in the tables referenced in Q2 and Q4, “100M” refers to the parameter budget in Q2, while it is interpreted as disk usage in Q4. Additionally, there are typos in the captions of both tables, “Methos.”
> > > >
> > > > Furthermore, I disagree with the authors’ approach of applying a threshold to singular values to retain only the leading singular vectors, respectfully. First, small singular values can also play an important role and should not be overlooked. Second, as the authors themselves mention, “many singular values are near-zero but not exactly zero.” If the authors choose to filter out small singular values, a more effective way to store these weights would be by using sparse matrix storage methods instead of SVD.
> > > > Third, using SVD introduces two sources of accuracy loss: the initial loss from filtering small singular values, and additional loss due to the rank constraints imposed by SVD. Therefore, sparse matrix storage would be a more suitable approach.

---

> > > > > ### Author Response · Authors · 2024-11-28
> > > > >
> > > > > ### **Q1: How to calculate the parameter budgets.**
> > > > >
> > > > > We calculate the XBGLoRA parameter budget as follows:
> > > > > - $m$ and $n$ is the input and output dimension of LoRA
> > > > > - $T$ is the number of LoRA boosters we use in Gradient Boost
> > > > > - $r$ is the rank of LoRA
> > > > > - $L$ is the number layers to which we add LoRA adapter
> > > > >
> > > > > Specifically:
> > > > > - For LoRA $T=1$, the parameter count is $L\times (m+n) \times r$.
> > > > > - For XGBLoRA $r=1$, the parameter count is $L\times (m+n) \times T$.
> > > > >
> > > > > In the previous response we set $r = T = 16$. Thus, the number of parameters of XGBLoRA and LoRA are equal:
> > > > >
> > > > > ($L\times (m+n) \times T = L\times (m+n) \times r) \approx 50M$).
> > > > > \
> > > > > \
> > > > > Since the precision of each parameter is float point 16bits (2 bytes),  50M parameters used 100 MB.

---

> ### Author Response · Authors · 2024-11-28
>
> ### **Q2: Sparsity vs. low-rank in the context of disk storage.**
>
> We appreciate and sincerely thank the reviewer's thoughtful suggestions regarding matrix storage methods. This perspective raises interesting questions about the relationship between sparsity and low-rank approximations.
> \
> \
> We would like to respectfully clarify some important technical points:
>
> - While we agree that efficient storage is valuable, we would like to clarify why SVD-based approaches provide theoretical and practical advantages in our context.
> - The key insight is that **sparsity and low-rank structure are fundamentally different matrix properties**, each with distinct implications for storage and approximation quality.
>
> Below, we provide a detailed analysis to explain why SVD-based approaches are more suitable for our specific use case:
>
> - The suggestion to use the sparse matrix storage instead of SVD appears to stem from perhaps misunderstanding of the mathematical properties of low-rank and sparse matrices. These are distinct concepts in linear algebra. Let us demonstrate this with two examples:
>    - A low-rank matrix that is completely dense:
>
>      $M_1 = \begin{pmatrix}
>      2.0 & 4.0 & 6.0 \\\\
>      3.0 & 6.0 & 9.0 \\\\
>      4.0 & 8.0 & 12.0
>      \end{pmatrix} = \begin{pmatrix}
>      2.0 \\\\
>      3.0 \\\\
>      4.0
>      \end{pmatrix} \begin{pmatrix}
>      1.0 & 2.0 & 3.0
>      \end{pmatrix}$
>
>    - A sparse but full-rank matrix:
>
>      $M_2 = \begin{pmatrix}
>      5.0 & 0.0 & 1.0 \\\\
>      0.0 & 4.0 & 2.0 \\\\
>      2.0 & 0.0 & 3.0
>      \end{pmatrix}$
>
>
> Kindly observe the following theorem:
>
> - **Theorem (Conditions for Sparsity in Matrix Factorization):**
> Given a matrix factorization $\mathbf{M} = \mathbf{AB}$ where $\mathbf{A} \in \mathbb{R}^{m \times r}$ and $\mathbf{B} \in \mathbb{R}^{r \times n}$, an entry $M_{ij} = 0$ if and only if one of these conditions holds:
>
>   - Zero-substructure condition: Either row i of $\mathbf{A}$ is zero or column j of $\mathbf{B}$ is zero
>   - Linear dependency condition: The vectors formed by row i of A and column j of B are orthogonal, i.e., $\langle A_{i:}, B_{:j} \rangle = 0$
>
> In our case, **these two conditions are unlikely to hold because SVD has no zero basis**, and the given matrix is non-symmetric so the second condition does not hold.
> \
> \
> Moreover, **common approaches to approximate dense matrices using sparse storage introduce larger errors than SVD:**
> \
> \
>   **a) Thresholding Method (zeroing elements below threshold τ):**
>
>   $M_{sparse} = \begin{pmatrix}
>   0.0 & 0.0 & 6.0 \\\\
>   0.0 & 6.0 & 9.0 \\\\
>   0.0 & 8.0 & 12.0
>   \end{pmatrix}$
>
>   **b) Random Dropping (randomly zeroing elements with probability p):**
>
>   $M_{random} = \begin{pmatrix}
>   2.0 & 0.0 & 6.0 \\\\
>   0.0 & 6.0 & 9.0 \\\\
>   4.0 & 8.0 & 0.0
>   \end{pmatrix}$
>
> **Both above methods have significant drawbacks:**
>
> - Larger error bounds: $\lVert \mathbf{M}\_1 - \mathbf{M}\_{sparse}\rVert\_F \\approx 5.385$ for thresholding
> - Random Error Norm: $[\lVert \mathbf{M}\_1 - \mathbf{M}\_{random}\rVert_F] \\approx 9.46$ for random dropping. For random dropping with probability $p$, the expected error is $E[\lVert \mathbf{M}\_1 - \mathbf{M}\_{random}\rVert_F] = p\lVert \mathbf{M}\_1\rVert_F$.
> - No guarantee of preserving matrix structure
> - Treat elements independently, ignoring global patterns
>
>
> In contrast, **The Eckart-Young-Mirsky theorem proves that truncated SVD provides the optimal low-rank approximation with minimal error:**
> - $\lVert \mathbf{M} - \mathbf{M}\_{svd}\rVert_F \leq \lVert\mathbf{M} - \mathbf{A}\rVert_F$ for any rank-k approximation $\mathbf{A}$, i.e., $\operatorname{rank}(A)=k$ and $\mathbf{M}\_{svd}=\sum\_{i=1}^k\sigma_i\mathbf{u}\_i\mathbf{v}\_i^T$ .
>
> **This is precisely why other successful methods like LoRA also employ low-rank approximations.**
>
> Concluding:
> - Our experimental comparisons with LoRA demonstrate that even when both methods use matrix factorization for storage efficiency, our approach achieves competitive performance while maintaining theoretical optimality guarantees.
>
> - We have validated these theoretical advantages through our comprehensive empirical evaluation, which demonstrates both storage efficiency and strong performance across various tasks. Kindly notice SBS strategy (no SVD) gave us 64.52\% and CDS variants (SVD-based) gave us 64.48\% and 64.50\%. **The variations are $\leq0.04$\% which are statistically irrelevant small variations.**
>
> - Finally, while storage is important, it is nowhere nearly as important as the GPU memory usage, *e.g.* 1TB SSD costs 100\\$, A100 40GB RAM GPU costs 15,000\\$, RTX4090 24GB RAM costs 2,000\\$ so **downscaling the GPU memory footprint is more essential than the storage**. We showed that **XGBLoRA enjoys $\mathcal{O}(1)$ memory complexity for rank-1 boosters**, and **LoRA requires  $\mathcal{O}(r)$ memory complexity**. Thus, we believe our method provides valuable advantages over LoRA.

---

> > ### Author Response · Authors · 2024-12-03
> >
> > Esteemed Reviewer,
> > \
> > \
> > As discussions are coming to an end, kindly do let us know if you would like us to include some additional clarifications in the manuscript revision apart from details from our discussion. We will follow your suggestions accordingly.
> > \
> > \
> > Best regards,
> > \
> > Authors

---

### Official Review · Reviewer_9zVu · 2024-11-02

**Soundness:** 3
**Presentation:** 3
**Contribution:** 3
**Rating:** 6
**Confidence:** 4

**Summary:**

This paper introduces a novel, efficient fine-tuning method for LLMs, XGBLoRA, inspired by gradient boosting. XGBLoRA randomly samples a subset of layers for rank-1 adaptation at each training step, significantly reducing the parameters required per step. XGBLoRA outperforms standard LoRA while using less GPU memory.

**Strengths:**

- The paper is clear and easy to understand, effectively explaining the ideas.
- The paper also presents a straightforward, efficient fine-tuning method that achieves state-of-the-art results with fewer parameters across various benchmarks.
- The theoretical analysis provided by the authors supports the method's expressiveness power.

**Weaknesses:**

- A significant limitation of the paper is that the reduction in trainable parameters does not significantly decrease GPU memory usage and training time. Furthermore, the frequent merging operations compromise LoRA's plug-and-play nature (efficient model storage), making the "Params" axis in Figure 1 misleadingly optimistic. Specific limitations include:
  - In Table 1, the computational cost for the base model $\beta$ appears higher than $\alpha$, suggesting that XGBLoRA does not significantly reduce the total cost. As shown in Figure 5, the difference in wall-clock time between 0.64s and 0.62s is minimal.
  - The optimization of an unstable subset of parameters in XGBLoRA adds complexity to the training pipeline, potentially slowing convergence. It would be beneficial to compare different total training steps $K$.

- There are minor errors in the Figures and Tables:
  - In Figure 4, the x-axis should be labeled $\kappa$ instead of $K$.
  - In Figure 5, the x-axis should be labeled as XGBLoRA, not GBLoRA.
  - In Table 5, bolded entries under Other, SIQA, and ARC-e columns are not optimal.
  - In Table 6, bolded entries under Social and Other columns are not optimal.

**Questions:**

- **Related work**: The paper omits a significant related work, COLA [r1], which also employs residual learning. Please provide a brief comparison of the key methodological differences between XGBLoRA and COLA, as well as to include COLA in their experimental comparisons if feasible.

    [r1] Xia, Wenhan, Chengwei Qin, and Elad Hazan. "Chain of lora: Efficient fine-tuning of language models via residual learning."

- **Theoretical justification**: The paper mentions that for LoRA, "To fit any target matrix, the rank of the adaptation must satisfy (r ≥ embedding_size/2)." For XGBLoRA, the paper presents Theorem 2. Please provide a more detailed explanation of how Theorem 2 demonstrates the expressiveness advantages of XGBLoRA over LoRA, particularly in relation to the trade-off between rank r and iterations T. And how to ensure consistent total training times between XGBLoRA and LoRA in the theoretical analysis.


Post-Rebuttal:

After reviewing author responses, all concerns are addressed.

---

> ### Author Response · Authors · 2024-11-23
> **Rebuttal for Reviewer 9zVu (part 1)**
>
> # Reviewer 9zVu
>
> Esteemed Reviewer,
> \
> We thank you for your valuable comments and constructive reviews that help us improve our work.
>
>
> ### **1. The reduction in trainable parameters does not significantly decrease GPU memory usage and training time.**
>
>
> The memory consumption for Parameter-Efficient Fine-Tuning (PEFT) of large language models can be broken down into four main components (using LLaMA-8B with LoRA rank 16 as an example):
>
> - **Base Model Parameters (~12GB)**: Required for storing the frozen pre-trained weights
> - **Forward Activation Memory (~10GB)**: Necessary for storing intermediate activations during forward pass
> - **Gradient Memory During Backward Pass (~1GB)**: Only required for adapter parameters in PEFT methods
> - **Optimizer State (~2GB)**: AdamW requires storage for both current and momentum states
>
> It is important to note that:
> - **ANY** PEFT method, including XGBLoRA, cannot reduce memory requirements for components **(1) and (2)** as they are fundamental to the training process.
> - As these components consume the majority of GPU memory (**approximately 22GB out of 25GB total**), any improvements in adapter efficiency may appear Non-significant in terms of total memory reduction. **Notice this is a case for any PEFT method**.
>
> However, **XGBLoRA makes significant improvements where possible compare to LoRA**:
>
> - **Uses rank-1 updates that require only 1/16th of the memory compared to standard LoRA (rank=16)**
> - **Optimizes components (3) and (4)**, which although smaller, are crucial for enabling deployment on more accessible hardware
> - **Most importantly**, this design is particularly significant as **XGBLoRA enables fine-tuning of large language models on consumer-grade hardware such as  the NVIDIA RTX 4090 (24GB)** instead of server grade A100, making LLM fine-tuning more accessible to a broader range of researchers and practitioners.
>
>
> ### **2. Frequent merging operations compromise LoRA's plug-and-play nature (efficient model storage).**
>
> 2.1. Thank you. We believe there may be some misunderstanding:
> - Reviewer might have assumed that XGBLoRA needs to store the entire model $\mathbf{W}{post}$ after training. This is not the case.
> - **XGBLoRA only need to maintain the final adapter weights $\Delta \mathbf{W}$**, which are significantly smaller than $\mathbf{W}{post}$.
> - These adapter weights are computed as:
> $$\Delta \mathbf{W} = \mathbf{W}{post} - \mathbf{W}{base}$$
>
> **This strategy lets XGBLoRA maintain its plug-and-play functionality, enabling the base model and adapters to be loaded independently.**
>
> 2.2. There are **two efficient storage strategies that maintain PEFT's plug-and-play nature** by allowing separately loading of the base mode $lW_{base}$ and adapter weights $\Delta W$:
>
> - **Compressed delta storage (CDS).**:
>
>   - Instead of storing $W_{post}$,  $\Delta W$ is compute and save as
>
>     $\Delta W = W_{post} - W_{base}$,
>
>     where $\Delta W$ is much smaller than the $W_{base}$.
>
>   - Since $\Delta W = \sum_{t=1}^{T} \Delta W^{(t)}$ is the sum of at most $T$ rank-1 matrices from boosting iterations, it remains low-rank.
>
>   - Therefore, we can further compress $\Delta W$ using SVD decomposition:
>
>     $\Delta W = U \Lambda V$,
>
>     and store only $A =  U\Lambda$ and $B = V$. This approach uses similar disk space as higher-rank LoRA but benefits from XGBLoRA's more efficient training process.
>
> - **Sequential Booster Storage. (SBS)**
>
>     - The alternative approach is to store individual rank-1 boosters ($A^{(t)}, B^{(t)}$) from each gradient boosting iteration and recover $\Delta W = \sum_{t=1}^T A^{(t)} B^{(t)}$ when loaded.
>
>     -  This results in a storage complexity of $O(T)$ for XGBLoRA (storing $T$ iterations of rank-1 matrices) compared to $O(r)$ for LoRA's rank-r adaptation.
>
>     -  Based on the therom 2 (as shown in Fig.2), XGBLoRA is able to achieve similar expressive error as LoRA,  where iteration $T$ is equal to LoRA rank $r$. Thus, $O(T)$ \approx O(r)$. SBS in XGBLoRA take simlar disk usage as LoRA.
>
>     -  In practice, since XGBLoRA converges in a limited number of iterations, this approach results in similar storage requirements to LoRA while maintaining lower GPU memory usage during training.
>
> 2.3. **We also provide the real disk usage** of two strategies above compared to **LoRA with rank 16** and **XGBLoRA with iteration 16**. Specifically, we first store and load the XGBLoRA via above two strategies and and evluate them in MMLU benchmark. Results are below:
>
> |Methos|DiskUsage|MMLU|
> |-|-|-|
> |LoRA (r=16)|~170M|64.01|
> |XGBLoRA (T=16)+CDS|~200M|64.50|
> |**XGBLoRA (T=16)+SBS**|~170M|64.52|

---

> ### Author Response · Authors · 2024-11-23
> **Rebuttal for Reviewer 9zVu (part 2)**
>
> ### **3. The optimization of an unstable subset of parameters in XGBLoRA adds complexity to the training pipeline, potentially slowing convergence. Compare to different total training steps.**
>
> Thank you.
>
> In the paper, we finetuned the model for 3 epcohs (each epoch has 335 traning steps) with LLaMA3-8B in the MMLU benchmark. To address the revewer's concerns, below we vary the traning epochs and report the average result as per request:
>
> |Epoch|1|3|5|7|9|
> |-|-|-|-|-|-|
> |LoRA|63.87|64.24|63.81|63.34|62.97|
> |**XGBLoRA**|64.62|65.33|65.19|65.27|**65.30**|
>
> - The introduction of randomness (referred to by rev. as *unstable subset of parameters*), while adding some complexity to fine-tuning process, follows established practices in ensemble learning to make model robust (**similar to dropout** in neural networks or **random splits in decision trees**). Such strategy aligns with **the gradient-boosting principle of creating weak but diverse learners.**
>
> - **For the convergence, even with just one epoch, XGBLoRA outperforms LoRA**, demonstrating efficient learning from limited fine-tuning steps.
>
> - **LoRA shows performance degradation with increased fine-tuning** (**dropping from 64.24 to 62.97** in the table above)
>
> - In contrast, **XGBLoRA maintains stable performance (around 65.30) across extended fine-tuning periods**.
>
> **The stable performance of XGBLoRA across different fine-tuning durations demonstrates the inherent robustness of gradient boosting algorithms**, which benefit from progressive refinement and the weak learner principle (highlighted in Sec 4.1)
>
>
> ### **4. Compare with COLA.**
>
> Thank you for highlighting the connection to CoLA.
>
> While we acknowledge CoLA (*currently an unpublished work-in-progress manuscript on arXiv*) uses a chain of LoRA adaptations that bears some resemblance to XGBLoRA's gradient boosting process, **there are fundamental differences both in theoretical foundation and technical implementation:**
>
> **4.1. Theoretical Framework:**
> - **XGBLoRA establishes a provable and interpretable theoretical framework** through the lens of **gradient boosting**
> - **Our analysis (Theorems 1 and 2)** explicitly **connects LoRA rank and the number of gradient boosting iterations to expressiveness error**, demonstrating how iterations can compensate for low rank
> - This theoretical framework provides **clear explanations for why iteratively merging LoRA adaptations benefits performance**
> - In contrast, **CoLA does not provide such theoretical justification** for its approach
>
> **4.2. Technical Implementation:**
> - **XGBLoRA explicitly adopts rank-1 adaptation and randomly selects layers for adaptation to limit adapter capacity.**
> - This design directly follows the weak learner principle from our gradient-boosting theoretical framework, where  strong ensemble model are built from diverse weak predictors
> - These specific technical choices are grounded in our theoretical analysis
> - In contrast, **CoLA does not incorporate these design elements (e.g. they use higher ranks and do not drop layers)**
> - **CoLA does not provide theoretical motivation or analysis for its architectural choices**
>
> **Our XGBLoRA provides both theoretical understanding and practical implementation strategies that are missing from CoLA's current formulation.** The theoretical foundations of XGBLoRA explain why and how the method works, while informing specific technical decisions that improve its performance where COLA does not.
>
> Below are the experiments as request :
> |        |MNLI   |SST-2 |CoLA   | QQP  | QNLI | RTE   | MRPC  |STS-B |Avg. |
> |---     |---    |---   |---    |---   |---   |---    |---    |---   |---  |
> |COLA    |88.06  |95.35 |64.70  |90.47 | 93.08| 84.62 | 89.63 |90.52 |87.05|
> |**XGBLoRA** |87.91  |95.70 | 66.28 |91.04 | 93.36 |86.10 | 90.57 |91.84 |**87.85**|
>
> **We will make sure to cite and acknowledge how CoLA and XGBLoRA relate/differ.**

---

> ### Author Response · Authors · 2024-11-23
> **Rebuttal for Reviewer 9zVu (part 3)**
>
> ### **5. Theoretical justification for Theorem 2.**
>
> Thank you.
>
> **Let us help analyze this mathematical comparison between LoRA and XGBLoRA using Theorem 2 and Fig. 2 in the paper** :
>
> **5.1. Model Characteristics.**
>
> - Note that:
>   - **LoRA** uses high rank $(r \gg 1)$ with single iteration $(T = 1)$
>   - **XGBLoRA** uses single rank $(r = 1)$ with many iterations $(T \gg 1)$
>
> - In Fig.2 we plot the theoretical error of XGBLoRA and LoRA:
>     - **XGBLoRA**: $error \leq C(1 + 1/\sqrt{T})$
>     - **LoRA**: $error ≤ C(1/r + 1)$, where $C$ is some shared constant just scaling both errors.
>
> - Fig.2 shows XGBLoRA can achieve comparable error bounds to LoRA by compensating for low-rank updates ($r=1$) with gradient boosting iterations ($T$) **while requiring only $\mathcal{O}(1)$ memory instead of LoRA's $\mathcal{O}(r)$.**
>
> **5.2. Convergence Analysis.**
> - XGBLoRA's error term $1/\sqrt{T}$ decreases more slowly than LoRA's $1/r$ as $T$ and $r$ increase respectively.
> - **This slower convergence actually indicates better regularization properties.**
> - **The slower decay suggests XGBLoRA is more resistant to overfitting compared to LoRA.**
>
> **5.3. Trade off.**
> The trade-off here is interesting:
>  - **XGBLoRA compensates for using lower rank ($r=1$)** by performing multiple boosting iterations ($T$), achieving comparable performance **while using much much less memory for fine-tuning.**
> - **The slower convergence rate actually becomes an advantage in terms of regularization** making model enjoy better performance
>
>
> ### **6. How to ensure consistent total training times between XGBLoRA and LoRA in the theoretical analysis.**
> In theoretical analysis, we simply assumed that LoRA use the same amount of time as XGBLoRA:
> - Suppose XGBLoRA has $T$ iteration and each iteration is train for $\kappa$ step.
> - The total number of training step for LoRA is $K = \kappa T$.
> - In the experiment, we fix the $K$ for both XGBLoRA and LoRA.
>
> **Therefore, XGBLoRA and LoRA received same amount time to fine-tuning the model.**

---

> > ### Comment · Reviewer_9zVu · 2024-11-25
> >
> > Thank you for your detailed response. Some of my concerns have been addressed. However, I still have a major question that seems to indicate a potential contradiction. In your manuscript, you mentioned that in each iteration, some layers are randomly selected for training, and the added LoRA is rank-1.
> >
> > - **When the number of iterations increases and the randomly selected $L_s$ is relatively large, XGBLoRA may face increasing storage and time costs**, requiring the use of difference computation for saving storage. In this case, even with the **CDS** and **SBS** strategies you mentioned, the **plug-and-play functionality** could still be compromised. Furthermore, regarding these two storage strategies, I suggest the authors elaborate on their implementation and effects in the updated manuscript.
> > - **When the number of iterations decreases and the randomly selected $L_s$ is small, the total trained parameters may become very small or even incomplete**. In this scenario, as you reported, XGBLoRA still outperforms LoRA. This is likely because the task itself does not require a large rank. Even with epoch=1, XGBLoRA (64.62) performs better than LoRA (63.87), which may not necessarily relate to "ensemble learning makes the model robust."
> > - I understand that XGBLoRA aims to strike a balance between them, and XGBLoRA will perform well on many simpler tasks. However, for unknown tasks, **a fixed number of iterations may not be sufficient, especially for more challenging tasks that do not converge as quickly as datasets like MMLU**. In contrast, LoRA’s fixed rank allows it to achieve better results with additional iterations. As noted above, increasing the number of iterations in XGBLoRA significantly exacerbates storage requirements.
> >
> > Additionally, I would like to know whether the learning rates used for LoRA and XGBLoRA were the same during comparison. Since **the two training strategies might require different optimal learning rates**, this could affect the results. I hope the authors carefully address the issues mentioned above, as they are critical to my scoring decision.

---

> > > ### Author Response · Authors · 2024-11-25
> > >
> > > Esteemed Reviewer,
> > > \
> > > \
> > > Rest assured we are working on a response plus experiments to respond to your further questions very soon.
> > > \
> > > \
> > > Best regards,
> > > \
> > > Authors

---

> > > ### Author Response · Authors · 2024-11-26
> > >
> > > ### **Q6. In contrast, LoRA’s fixed rank allows it to achieve better results with additional iterations. As noted above, increasing the number of iterations in XGBLoRA significantly exacerbates storage requirements.**
> > >
> > > Kindly notice **this is also true for XGBLoRA.**
> > >
> > > If the total training step $K$ is increases under the fixed iteration $T$, **the training interval $\kappa$ for each booster before merging also increases since $\kappa = \frac{K}{T}$.**
> > > \
> > > \
> > > **This means we can train each booster longer before merging (same as for reviewer's argument about LoRA) and this can increase the capacity of the LoRA booster for fitting the data.**
> > > \
> > > \
> > > Moreover, although in our paper we use rank-1 as the weaker learner for GB due to the simple task, the ability of the weak learner should be chosen relative to the complexity of the task. **Thus, for complex tasks, our booster may use rank higher than 1, *e.g.,* it could be rank-2, rank-3 to match the match the task complexity.**
> > >
> > > Choosing the booster's capacity is the common practice in the GB algorithms. For example, in the gradient boost decision tree, people might slightly increase the tree depth for different datasets rather than stick to tree stump (tree with 1 split).
> > >
> > > **Should reviewer further require us to vary intervals for LoRA and XGBLoRA** to demonstrate their effect, we are more than happy to run such an experiment. We expect both  LoRA and XGBLoRA to behave according to reviewer's intuition here (better results with additional iteration for more complex data).
> > >
> > >
> > > ### **Q7. Additionally, I would like to know whether the learning rates used for LoRA and XGBLoRA were the same during comparison.**
> > >
> > > Thank you.
> > >
> > > We use the same learning rate 1e-4 (the learning rate in backpropagation) for XGBLoRA and LoRA. This is because LoRA is a special case of XGBLoRA with  $T=1$. We try to make them consistent in evaluations. Nonetheless, below is the effect of changing LR for LoRA and XGBLoRA. We fine-tune on LLaMA3-8B and evaluate on MMLU benchmark.
> > >
> > > | LR            |  5e-5   |  1e-4   |  5e-4   |
> > > |---|---|---|---|
> > > | XGBLoRA       |  65.32  |  65.37 |  65.29   |
> > > | LoRA          |  64.08  |  64.10 |  64.06   |
> > >
> > > According to the table, changing LR has marginal impact on both methods.
> > >
> > > If reviewer asked about the learning rate of the gradient boost $\alpha$ in Eq. 8, it is always set to $\alpha=1$ in our experiments.

---

> > > > ### Comment · Reviewer_9zVu · 2024-12-01
> > > >
> > > > Dear Authors,
> > > >
> > > > I sincerely appreciate the author's thoughtful and detailed rebuttal. Most of my concerns have been addressed. While I believe this method demonstrates improvements over LoRA (e.g., XGBLoRA may overfit less to simple dataset), the paper somewhat overstates its performance. On simple datasets, the comparison with the baseline does not require larger ranks and epochs; however, the manuscript does not reflect this, which to some extent underestimates the baseline. Therefore, I am inclined to maintain my original score.

---

> ### Author Response · Authors · 2024-11-26
>
> ### **Q1. When the number of iterations increases and the randomly selected $L_s$ is relatively large, XGBLoRA may face increasing storage and time costs, requiring the use of difference computation for saving storage. Even with the CDS and SBS strategies you mentioned, the plug-and-play functionality could still be compromised.**
>
> Thank you.
>
>  - We agree that CDS and SBS may not be compact if the number of iterations is large.
>    - However, for CDS, XGBLoRA storage is bounded by the maximum $\bf\Delta W$ matrix rank, and the LoRA's storage is also bounded by LORA's $\bf\Delta W$ matrix rank.
>    - In contrast, the size of SBS  depends strictly on $T$
>
> However, **we would like to humbly highlight that in the context of parameter-efficient fine-tuning (PEFT), disk storage is less critical than the GPU memory usage** because in most cases PEFT researchers are constrained by the limited GPU memory rather than disk storage (1TB SSD costs \\$100, Tesla A100 with 40GB costs over \\$15,000 while RTX4090 with 24GB memory costs \\$2,000). **Here, our method has a favorable GPU memory usage $\mathcal{O}(1)$ under rank-1 boosters versus $\mathcal{O}(r)$ for LoRA.**
>
> In conclusion:
> - The plug-and-play functionality could only be compromised when $T\gg r$. However, **based our Theorem 2 (kindly see Fig. 2), XGBLoRA achieves similar expression error as LoRA when $T=r$.**
> - Please note the small gap between the red and black curve (Fig. 2) for large value of x-axis showing effect of rank $r$ and iteration $T$ are similar. **Therefore, based on the theoretical error analysis we do not expect $T$ of XGBLoRA to be much larger than $r$ rank of LoRA.**
> - Notice **we can further lower that gap in Fig. 2 if needed by increasing the rank of our weak booster from 1 to 2 or so.**
> - Thus, plug-and-play functionality should be preserved in practice. **We also have not observed any issues with the plug-and-play functionality/disk storage on datasets we worked on. In contrast, we observed out-of-memory GPU errors for LoRA.**
>
> ### **Q2. I suggest the authors elaborate on their implementation and effects in the updated manuscript.**
>
> Absolutely. We will make sure to add this discussion to our revised manuscript. We also wish readers to be fully aware of various limitations and constraints of XGBLoRA.
>
> ### **Q3. This is likely because the task itself does not require a large rank.**
>
> We believe reviewer is correct.
>
> For simple datasets, the key consideration in the fine-tuning step is to avoid overfitting. This ability is expected from LoRA, XGBLoRA and any other PEFT method.
> \
> \
> **The part of the ability of XGBLoRA stems from the implicit regularization that is inherent to the gradient boost mechanism** (this is what gradient boosting does by design).
> \
> \
> Preventing overfitting requires a regularization mechanism **(a.k.a. the prior belief) and not all regularization mechanisms are equally good:**
> - **XGBLoRA can be controlled by:**
>   - the number of boosters
>   - rank of individual boosters
>   - and even number of steps on each booster (similar to early stopping in SGD optimized losses)
>   - layer selection
> - **LoRA can control overfitting only by:**
>    - number of steps
>    - rank $r$
>
> Thus, XGBLoRA may overfit less to simple (low number of samples) dataset, e.g., this is what we observed on the the Alpaca dataset (very small dataset, mere ~1k instructions).
>
>
> ### **Q4. Even with epoch=1, XGBLoRA (64.62) performs better than LoRA (63.87), which may not necessarily relate to "ensemble learning makes the model robust."**
>
> To further verify robustness of regularization effect in the ensemble learning in XGBLoRA, we compare it with LoRA of various ranks:
>
>   |Method| MMLU Performance|
>   |---|---|
>   |LoRA (r=1)     |**64.8**|
>   |LoRA (r=4)     |64.33|
>   |LoRA (r=8)     |64.10|
>   |LoRA (r=16)     |63.51|
>   |XGBLoRA        |**65.3**|
>
> We obtain the above results via fine-tuning LLaMA-8B on the Alpaca (en) dateset and evaluating on the MMLU benchmark. **As Alpaca (en) has mere ~1k instructions, it is prone to cause overfitting.**
> \
> \
> **The table shows that:**
> - LoRA (rank r=1) achieves the best regularization compared to higher LoRA ranks.
> - **XGBLoRA still outperforms LoRA (rank r=1)**, thus XGBLoRA exhibits better regularization properties.
>
>
> ### **Q5. For unknown tasks, a fixed number of iterations may not be sufficient, especially for more challenging tasks that do not converge as quickly as datasets like MMLU.**
>
> We understand reviewer may think complex task may require significant large iterations. Our Theorem 2 (explained in Fig. 2 in the paper) suggest it may not be the case as the gap of expression error between XGBLoRA and LoRA is relatively small under $T=r$.
>
> Notice that **we can further lower that gap in Fig. 2 if needed by increasing the rank of our weak booster from 1 to 2 or so.**

---

> ### Author Response · Authors · 2024-12-01
>
> Esteemed Reviewer,
> \
> \
> We understand and value your concerns. **Everything we discussed here will be included in the manuscript - that is why we are discussing it here.**
> \
> \
> **If reviewer feels some aspects of our work are overstated, we respectfully ask if reviewer can concretely tell us what we should modify and how.** We just report performance based on experiments and concrete numbers we achieved: we do not understand what it means we **overstate**  XGBLoRA's performance. We just show empirical evidence backed by the theory.
> \
> \
> At this point we feel we have replied to reviewer's all criticisms in good faith that our efforts will not be ignored, and we are happy to address more concerns if we are concretely told what more is expected of us.
> \
> \
> **Our work explains theoretically  how iterations $T$ and tank $r$ interact and impact the error. This alone is a valuable contribution for a conference that values theoretical works.** Thus, we struggle to understand what more we can do to convince the reviewer.
> \
> \
> Kind regards,
> \
> Authors

---

> ### Comment · Reviewer_9zVu · 2024-12-01
>
> Esteemed Authors,
>
> I sincerely appreciate your detailed response. In the manuscript, the default rank for LoRA is set to 8, but based on the rebuttal, I find that some datasets do not require such a high rank (in some cases, rank-1 is optimal). As a result, the significant reduction in Params presented in the manuscript may be somewhat overemphasized. Similarly, regarding the theoretical explanation, the manuscript states that “Original LoRA is the case where T = 1 and r ≫ 1,” which could be slightly overstated. A more accurate phrasing would be “r ≥ 1.”
>
> Additionally, in the rebuttal experiments, when epoch=1, XGBLoRA outperforms the optimal LoRA. In my view, XGBLoRA at this point likely trains only a subset of LoRA parameters and does not fully utilize ensemble learning, which seems somewhat abnormal. Furthermore, this point was not adequately clarified in your reply to Q4.
>
> I acknowledge that incorporating multiple LoRA layers through ensemble learning is a noteworthy improvement. I look forward to future iterations of the manuscript that include an ablation study on the rank of LoRA and the number of iterations K to further validate the effectiveness of XGBLoRA.
>
> Kind regards,
>
> Reviewer

---

> > ### Author Response · Authors · 2024-12-01
> >
> > Thank you,
> > \
> > \
> > From the reviewer's comments we believe we are on the same page with the reviewer's points. Absolutely, LoRA is $T=1$ and $r\geq 1$. We are more than happy to correct it so.
> > \
> > \
> > We are also working on clarifying Q4 as per reviewer's request and further ablations. We will provide more results as soon as experiments complete.
> > \
> > \
> > Kind regards,
> > \
> > Authors.

---

> ### Author Response · Authors · 2024-12-02
>
> Esteemed Reviewer,
> \
> \
> We apologize for the delay in experimentation (some authors were in travel). Now we have more results which hopefully clarify further the issues raised. **We kindly ask Reviewer to tell us how Reviewer would like us to present these results in the revised manuscript, and we will do so accordingly** in order to reflect reviewer's concerns precisely.
> \
> \
> ++++
>
> ### **Q1. I find that some datasets do not require such a high rank (in some cases, rank-1 is optimal). As a result, the significant reduction in Params presented in the manuscript may be somewhat overemphasized.**
>
>
> - **We acknowledge that certain datasets may not require high-rank adaptations.** When fine-tuning powerful models like LLaMA3-8B on simpler datasets, **overfitting become a significant concern**. In such cases, **the regularization capability becomes more valuable than parameter efficiency.** However, this limitation stems from the inherent characteristics of the dataset rather than our model (e.g., **the simple Apaca dataset has mere ~1k instructions and is not suitable for fine-tuning due to its extremely small size**).
>
> - Thus, we highlight that XGBLoRA demonstrates superior regularization capabilities compared to standard LoRA, even in rank-1 settings. This enhanced performance can be attributed to the robust nature of ensemble learning.
>
>
>    |Epoch	        | 1	  |   3	 |    5	 |     7  |
>    |---            |---  |---   |---    |---     |
>    |LoRA (r=1)	    |64.24|	**64.81**|	64.71|	64.64 |
>    |XGBLoRA	    |64.62|	**65.33**|	65.19|	65.27 |
>
>    The table shows that **under rank-1, varying the number of epochs for LoRA to obtain its best result does not outperform XGBLoRA.**
>
> ++++
> ### **Q2. Additionally, in the rebuttal experiments, when epoch=1, XGBLoRA outperforms the optimal LoRA. In my view, XGBLoRA at this point likely trains only a subset of LoRA parameters and does not fully utilize ensemble learning, which seems somewhat abnormal.**
>
> - **Reviewer may think we randomly selected the subset of layers per epoch in XGBLoRA, thus concluding that with 1 epoch training, XGBLoRA trains only a subset of LoRA parameters.**
>
> - However, **this is not true** as **we randomly select the new subset of layers per each GB iteration rather than per epoch**.
> - Each epoch of XGBLoRA contains multiple GB iterations. In the case of $\kappa = 8$, **each epoch has 46 GB iterations** so **we touch/train all layers** in a single epoch, just not in a single iteration.
>
> Thus, **XGBLoRA with 1-epoch training merges 46 randomly sampled subsets of LoRA parameters.** This aligns with **the fundamental concept of ensemble learning**, where strong models are built by combining diverse weak learners.
> \
> \
> Below we simulate the case we believe that reviewer suggests (kindly correct us if we misunderstood you).
> \
> **We randomly select the subset of LoRA layers once per epoch** for optimization and train the fixed subset of LoRA layers per epoch:
>
> |       One Epoch Training| MMLU  |
> |---                        |---    |
> |LoRA (r=1, **layer subset per epoch**)        | 64.02 |
> |LoRA (r=1, **layer full set (standard)**)        | **64.24** |
> |XGBLoRA (**layer subset per epoch**)        | 64.30 |
> |XGBLoRA (**layer subset per GB Iteration**) | **64.62** |
>
>
> **Kindly notice that when fine-tuning on the simple Apaca dataset:**
> - **selecting subset of LoRA layers in XGBLoRA per GB Iteration for optimization gives the best result** (all layers are **touched** per epoch)
> - **selecting subset of LoRA layers in XGBLoRA per epoch for optimization is visibly worse** (only **subset** of layers are touched per epoch)
> - **selecting subset of LoRA layers in LoRA (rank r=1) per epoch for optimization results in drop (64.02)** versus 64.24 for standard LoRA.
>
> We acknowledge not all parameters are optimized per one GB iteration, but all parameters are touched/optimized within an epoch.
> \
> \
> **If we have just a subset of parameters per epoch optimized, results in the table are worse**.  This highlights that ensemble learning plays a pivotal role in reducing overfitting.
> \
> \
> We hope we understood what the reviewer meant by **optimising a a subset of LoRA parameters** and we tried to show that this strategy does not benefit the standard LoRA. **We believe our finding does not contradict reviewer's intuition: it just additionally highlights the strength of ensemble learning.**

---

> > ### Comment · Reviewer_9zVu · 2024-12-03
> >
> > Thank you for your timely and thoughtful response. My concerns have mostly been addressed, and **I have increased the score to 6**. I would like to request the following modifications in the revised manuscript:
> >
> > - Please provide the results for LoRA at rank=1 to highlight that XGBLoRA demonstrates superior regularization capabilities.
> > - The manuscript does not clearly describe the concept of epochs; please include this.
> > - The theoretical section needs further elaboration about XGBLoRA's Expressiveness Error at r=1.
> >
> > Lastly, I still have a question: you mentioned that **each epoch has 46 GB iterations**. Does this imply that multiple epochs would result in more parameter updates? Could this lead to an increase in the rank of the update matrix, potentially causing storage pressure?

---

> > > ### Author Response · Authors · 2024-12-03
> > >
> > > We would like to thank the reviewer for a really thorough discussion that has helped us refine our work in many critical aspects.
> > >
> > > ====
> > > ### **Q1. I would like to request the following modifications in the revised manuscript**
> > >
> > > Absolutely. Rest assured we will follow your requested modifications accordingly.
> > >
> > > ====
> > > ### **Q2. Does this imply that multiple epochs would result in more parameter updates? Could this lead to an increase in the rank of the update matrix, potentially causing storage pressure?**
> > >
> > > Thank you. Using more iteration will result in more parameters updates and it will increase the required storage.
> > >
> > > - However, we can control the iteration $T$ via setting parameter $\kappa$ (the training step for each booster before merging). Suppose the parameter of total training steps $K$ in one epoch is fixed ($T = \frac{K}{\kappa}$), then we can increase the $\kappa$ parameter to reduce the iteration $T$.
> > >
> > > - By increasing $\kappa$, we can slightly increase capacity of booster as we train each booster with more steps, thus XGBLoRA may not need as many GB iterations to maintain its expressive power.
> > >
> > > - To demonstrate the above point, below we set the parameter updates of XGBLoRA to be the same as the parameter updates of LoRA by setting the GB iteration $T$ to be equal LoRA rank $r=16$.
> > >
> > >
> > > |Methods                 |#Parameter updates | MMLU|
> > > |---                     |---                |---  |
> > > |LoRA (r=16)             |~50M               |64.01|
> > > |XGBLoRA (T=16)          |**~50M**               |**64.52**|
> > > |XGBLoRA (T=46)          |~150M              |64.62|
> > >
> > > - As one can see from the table, reduced the parameter updates by reducing $T$ lead to a minimal performance drop (64.52 versus 64.62. **The 64.52 result uses the  parameter count of LoRA, which performed 64.01**)
> > >
> > > **We will provide the above ablation and extend it with more entries in the revised manuscript to give readers better and transparent idea of parameter-storage-result trade-offs.**
> > > \
> > > \
> > > Beside of ablations, we would like to humbly highlight that in the context of parameter-efficient fine-tuning (PEFT), disk storage is perhaps bit less critical than the GPU memory usage because in most cases PEFT researchers are constrained by the limited GPU memory rather than disk storage (1TB SSD costs \\$100, Tesla A100 with 40GB costs over \\$15,000 while RTX4090 with 24GB memory costs \\$2,000).
> > > \
> > > \
> > > Here, our method has a favorable GPU memory usage $\mathcal{O}(1)$ under rank-1 boosters versus for LoRA $\mathcal{O}(r)$. For us, being able to switch to RTX4090 due to GPU RAM savings was particularly beneficial in terms of cost savings.
> > > \
> > > \
> > > Once again, we thank the reviewer for a really thorough and valuable discussion.

---

### Official Review · Reviewer_rfLt · 2024-11-03

**Soundness:** 3
**Presentation:** 3
**Contribution:** 2
**Rating:** 5
**Confidence:** 3

**Summary:**

This paper gives a novel framework that enhances LoRA through ensemble learning, bringing it closer to theoretical optimality. The proposed method’s convergence is demonstrated through detailed theoretical analysis. Comprehensive experiments reveal the relationship between the number of weak learners and model performance. Additionally, XGBLoRA outperforms traditional LoRA and many baseline methods across multiple tasks while significantly reducing the number of trainable parameters and memory usage.

**Strengths:**

1. It is a good idea to use gradient boosting in LoRA finetuning, and the results showed that the idea is valid.

2. The concepts in the paper are clear and understandable. This paper has a well-organized structure that effectively conveys the authors’ ideas.

3. This paper provides detailed theoretical analysis and proofs.

**Weaknesses:**

1. In Lines 251-256, could frequent merging introduce additional overhead? This part lacks quantitative analysis.

2. Line 413 mentioned that the hyperparameters refer to LoRA[1], but to my knowledge, multiple sets of hyperparameters were provided, which seems ambiguous. Moreover, I am uncertain if the hyperparameters for RoBERTa and GPT-3 are suitable for the current models.

3. In Table 5, the performance improves as the rank decreases. Could this be due to the simplicity of the data, which fails to allow for sufficient fitting?

4. Alpaca is an excellent project; however, the dataset is relatively simple (despite being regenerated with GPT-4). Meanwhile, base models such as LLaMA3-8B and Mistral exhibit strong performance. Based on my experience, fine-tuning a highly capable base model with a lower-quality dataset can lead to model degradation.

	(a) From the existing leaderboards[2], LLaMA3-8B has achieved 66.49% accuracy on MMLU, it is even higher than the best value in Table 4. To avoid discrepancies due to evaluation methods and code versions, could you provide the performance of base models like LLaMA3-8B and Mistral on 8 tasks?

	(b) The Alpaca GPT-4(en) dataset differs significantly from the currently popular instruction fine-tuning datasets. Could you share results from other datasets like WizardLM[3], Infinity-Instruct [4] ?

[1] https://arxiv.org/pdf/2106.09685

[2] https://huggingface.co/spaces/open-llm-leaderboard-old/open_llm_leaderboard

[3] https://huggingface.co/datasets/WizardLMTeam/WizardLM_evol_instruct_70k

[4] https://huggingface.co/datasets/BAAI/Infinity-Instruct

**Questions:**

Please refer to Weaknesses.

---

> ### Comment · Reviewer_BBeS · 2024-11-23
> **Reviewer BBeS's Comment**
>
> I believe the weakness 3 raised by Reviewer rfLt can be explained effectively. In the PEFT domain, larger parameter sizes do not necessarily correspond to better performance. This viewpoint is supported by experimental results from most studies and has even been theoretically validated by certain works, such as the 2023 AAAI paper “On the Effectiveness of Parameter-Efficient Fine-Tuning.”
>
> However, Reviewer rfLt’s concern is highly insightful. I would like to ask whether the authors could provide a performance comparison between XGBLoRA and LoRA under the same parameter budget. Since larger parameter sizes do not necessarily guarantee higher performance, a comparison under identical parameter settings would be more meaningful.

---

> ### Author Response · Authors · 2024-11-23
> **Rebuttal for Reviewer rfLt**
>
> # Reviewer rfLt
>
> Esteemed Reviewer,
> \
> We thank you for your valuable comments and constructive reviews that help us improve our work.
>
> ### **1. Will frequent merging introduce addtion oevrhead**
>
> XGBLoRA's merging process incurs minimal computational overhead:
> - **Rather than unload and create new LoRA adapters at each gradient boosting iteration, we simply clear gradients and reinitialize parameters.**
> - **The merging operation itself is just tensor addition**, making its computational cost negligible compared to the model's forward and backward passes.
>
> - For **LLaMA3-8B (d=4096, k=4096) with $κ=8$ minibatches** between merges:
>   - Training cost per layer per minibatch: ~101M FLOPs
>   - Total training cost over κ minibatches: 8 × 32 × 101M ≈ 25.9B FLOPs
>   - **In contrast, one-time merging cost for Ls=8 layers for every $\kappa$ minibatches:  1 × 8 × 16.8M ≈ 134M FLOPs**
> - **This makes merging overhead only ~0.5% (134M/25.9B) of total computation between merges**, demonstrating that merging introduces negligible computational overhead in practice.
>
>
> ### **2. List the hyperparamter that we used for traning the LLAMA3-8B and Mistral-7B.**
>
> - For initialization, we followed the approach outlined in the LoRA paper, **where the $A$ matrix is initialized using a zero matrix and $B$ is initialized with Kaiming initialization.**
>
> - While we employed different hyperparameter sets for various benchmarks, we have now documented all these **details in a new section in the appendix (Kindly check the revision for details).**
>
> - an an example, follwoing are the hyperparamters used for commonsense reasoning tasks (Note that Rank $r$ is for LoRA family, **XGBLoRA use rank $1$ as default**):
> |Hyperparameters|LLaMA-7B|LLaMA-13B|LaMA3-8B|
> |-|-|-|-|
> |Rank r|8|8|8|
> |Dropout|0.05|0.05|0.05|
> |Optimizer|AdamW|AdamW|AdamW|
> |LR|2e-4|3e-4|1e-4|
> |LR Scheduler|Linear|Linear|Linear|
> |Batch size|1|1|1|
> |AccumulationSteps|4|4|4|
> |Warmup Step|100|100|100|
> |Epochs|3|3|3|
>
> Kindly note **plenty more hyper-parameters are now in Appendix**.
>
> - It is worth noting that the hyper-parameters unique to XGBLoRA (specifically $\kappa$ and $L_s$) remain consistent across all datasets and benchmarks.
>
>
> ### **3. Performance of base models like LLaMA3-8B and Mistral on 8 Tasks.**
>
> We test the base model (without fine-tuning). Results are below:
> |Methods|BoolQ |PIQA |SIQA |HellaSwag |WinoGrande |ARC-e |ARC-c |OBQA |Avg.|
> |-|-|-|-|-|-|-|-|-|-|
> |LLaMA3-8B (base)|72.8|84.2|70.5|79.3|74.6|86.1|76.4|74.0|77.2|
> |Mistral-7B (base)|72.1|83.5|65.6|75.5|73.7|82.8|68.3|72.4|74.2|
> |**XGBLoRA (fine-tune on LLaMA3-8B)**|72.5|85.8|78.3|93.5|83.9|88.1|75.1|84.2|83.0|
>
>
> ### **4. Results from other datasets like WizardLM [3], Infinity-Instruct [4].**
>
> Thank you for raising these important points about dataset quality and model evaluation.
> \
> \
> Your concerns about potential model degradation when fine-tuning capable base models on simpler datasets are valid.
>
>
> - Regarding alternative instruction tuning datasets, we conducted additional experiments on using WizardLM's evol_instruct dataset (~70K examples).
>
> |WizardLM Results|MMLU|
> |-|-|
> |LLaMA2-7B (base)|46.87|
> |LLaMA2-7B + LoRA|45.79|
> |LLaMA2-7B + XGBLoRA|**47.64**|
> |Mistral-7B (base)|59.11|
> |Mistral-7B + LoRA|58.20|
> |Mistral-7B + XGBLoRA|**60.42**|
>
> These results demonstrate several key points:
> - **XGBLoRA consistently improves performance over both the base model and LoRA across different model architectures**
> - **XGBLoRA manages to enhance performance**, while **LoRA shows some performance degradation** compared to the base models.

---

> ### Author Response · Authors · 2024-11-23
> **Re: Reviewer BBeS's Comment (XGBLoRA and LoRA under the same parameter budget).**
>
> Dear Reviewer BBeS,
> \
> \
> We sincerely thank the reviewer for helping us improve our work.
> \
> \
> We have now answered this question in **Rebuttal for Reviewer BBeS (part 3), Question 5.**
>
> Kind regards,
> \
> Authors.

---

> ### Comment · Reviewer_rfLt · 2024-11-24
> **Reviewer Feedback**
>
> Dear Authors,
>
> Thank you for providing the experiment details.
>
> I noticed in your response to point 4 that the results of the **Base model are higher than those of Base + LoRA (46.87 > 45.79, 59.11 > 58.20)**, which seems unconventional. Considering the result of mistral-7b-base you provided, I also observed that the improvement of XGBLoRA in Table 4 is minimal (59.26 > 59.11), and both **FT and LoRA perform worse than the Base model (59.11 > 58.99 > 58.09)**. Therefore, I am concerned that the **baseline might have been underestimated**.
>
> Additionally, I am still curious about the performance of llama3-8b on MMLU as evaluated by your codebase. If you could provide this information, I would greatly appreciate it.
>
> Best,
>
> Reviewer rfLt

---

> > ### Author Response · Authors · 2024-11-25
> >
> > Esteemed Reviewer,
> > \
> > \
> > Rest assured we are working on a response plus experiments to respond to your further questions very soon.
> > \
> > \
> > Best regards,
> > \
> > Authors

---

> ### Author Response · Authors · 2024-11-25
>
> ### **Q1. I am still curious about the performance of llama3-8b on MMLU as evaluated by your codebase.**
>
> To avoid further misunderstanding, we share the link of public codebase and Apaca(en) we used in our experiments:
>
> - codebase: https://github.com/hiyouga/LLaMA-Factory
> - Apaca(en): https://github.com/hiyouga/LLaMA-Factory/tree/main/data
>
> The Base model performance of LLaMA-8B on MMLU is 66.03 \%. The complete table (fine-tuning on Apaca (en)) is given below:
>
> |          |LLaMA3-8B |  Mistrl-7B|
> |---       |---       |---        |
> |Base      |  66.03   |   59.11   |
> |FT        |  63.26   |   58.09   |
> |LoRA (r=8)|  64.10   |   58.99   |
> |XGBLoRA   |  65.37   |   59.34   |
>
> ### **Q2. FT and LoRA perform worse than the Base model (59.11 > 58.99 > 58.09). Therefore, I am concerned that the baseline might have been underestimated.**
>
> - For LLaMMA-8b, the base model also performed the best.
>
>   The overall order of performance is as follows:
>
>   **Base > XGBLoRA > LoRA > FT**
>
> - **We believe the reviewer argument in the initial review is insightful and correct:**
>   >fine-tuning a highly capable base model with a lower-quality dataset can lead to model degradation
>
>
> - To be precise, we believe this degradation is not due to underestimating the baseline (FT and LoRA). Rather, it occurs because fine-tuning the powerful LLaMA model leads to overfitting on simple data, resulting in a poor generalization on downstream evaluation tasks (MMLU).
>
>   **To verify this, we vary the rank of LoRA:**
>   |Lora rank $r$| 1 | 4 | 8 | 16|
>   |- |---|---|---|---|
>   |LoRA     |**64.8**|64.33| 64.10| 63.51|
>
>   One see that **decreasing model complexity (reducing rank from 8 to 1)** improves the performance of LoRA, while **increasing complexity (rank 8 to 16) degrades the performance.**
>    This verifies our hypothesis that **model fine-tuned LoRA on LLaMA3-8B  overfits to the simple Apaca dataset (mere \~1k instructions) causing model degrade.**
>
> - To further verify the issue with overfitting, we vary the number of epochs which implicitly affects the model complexity:
>
>    | Number of epochs   |  1    | 3     | 5     |   7   |   9   |
>    |---|---|---|---|---|---|
>    |LoRA    | 63.87 | **64.24** | 63.81 | 63.34 | 62.97 |
>    |XGBLoRA | 64.62 | **65.33** | 65.19 | **65.27** | **65.30** |
>
>   When we increase the risk of overfitting by running more epochs during fine-tuning, **the performance of LoRA decreases from 64.24 (3 epochs) to 62.97 (9 epochs).**
>
>   However, **XGBLoRA maintains stable performance in the  across epochs due to ensemble learning's inherent resistance to overfitting.**
>
> - We agree with the reviewer that Alpaca (en) may be too simplistic (**\~1k instructions is not a lot**) for fine-tuning. Thus, following the reviewer's suggestion, **we have conducted additional experiments with a more complex dataset WizardLM (~70k instructions) using lower capacity models:**
>
> |WizardLM Results|MMLU|
> |-|-|
> |LLaMA2-7B (base)|46.87|
> |LLaMA2-7B + LoRA|45.79|
> |LLaMA2-7B + XGBLoRA|**47.64**|
> |Mistral-7B (base)|59.11|
> |Mistral-7B + LoRA|58.20|
> |Mistral-7B + XGBLoRA|**60.42**|
>
> We believe these results demonstrate XGBLoRA's superiority through two key mechanisms of GB:
>
> - **The Progressive refinement** - achieving optimal performance without explicitly using higher rank
> - **Weak learner principle** - maintaining robustness by combining multiple rank-1 booster
>
> We truly hope this can resolve the reviewer's concerns about the model degradation. Our observations are aligned with the reviewer's initial comment/observation. We believe Alpaca simply is not a suitable/large enough dataset for fine-tuning.

---

> > ### Comment · Reviewer_rfLt · 2024-11-30
> > **Reviewer Feedback II**
> >
> > I sincerely appreciate the author's genuine response. After some consideration, I decided to maintain my score. Because I believe it would be irresponsible to compare various fine-tuning methods when they cannot always bring consistent improvements.

---

> ### Author Response · Authors · 2024-11-30
>
> Esteemed Reviewer,
> \
> \
> We sincerely thank for your response and we value your feedback.
> \
> \
> However, our method does achieve consistent improvements over other PEFT methods. Among all, **even on Apaca we outperform other PEFT methods despite Apaca has mere 1k tuning instructions.**
> \
> \
> As such, **we believe this is the issue with Apaca just being too small for all PEFT methods** when they lag behind base, and we showed that consistently. We hope reviewer can sympathize it is not our fault that **this dataset is so small that it is simply not enough for parameter-efficient fine tuning.** The problem reminds us of classical overfitting when trying to fine-tune ResNet-101 with mere 1000 images...
> \
> \
> Kindly notice that on **WizardLM's instruct dataset with about 70K examples the improvements achieved by XGBLoRA are clear cut.**
> \
> \
> Best regards,
> \
> Authors

---

### Comment · Area_Chair_JWcc · 2024-11-25
**Interactive Discussions**

Dear Reviewers,

Thank you for your efforts in reviewing this paper. We highly encourage you to participate in interactive discussions with the authors before November 26, fostering a more dynamic exchange of ideas rather than a one-sided rebuttal.

Please feel free to share your thoughts and engage with the authors at your earliest convenience.

Thank you for your collaboration.

Best regards, ICLR 2025 Area Chair

---

### Note · Authors · 2025-01-31

I have read and agree with the venue's withdrawal policy on behalf of myself and my co-authors.